

# LIME: Lunar Irradiance Model of ESA, a new tool for the absolute radiometric calibration using the Moon

Carlos Toledano[1], Sarah Taylor[2], África Barreto[3], Stefan Adriaensen[4], Alberto Berjón[5,3],
Agnieszka Bialek[2], Ramiro González[1], Emma Woolliams[2], and Marc Bouvet[6]

[1]Group of Atmospheric Optics, University of Valladolid, 47011 Valladolid, Spain
[2]National Physical Laboratory, Hampton Road, Teddington, Middlesex, TW11 0LW, UK
[3]Izaña Atmospheric Research Center (IARC), State Meteorological Agency of Spain (AEMET), 38001 Santa Cruz de Tenerife, Spain
[4]Flemish Institute for Technological Research (VITO), Boeretang 200, 2400 Mol, Belgium
[5]TRAGSATEC, 28006 Madrid, Spain
[6]European Space Agency (ESTEC), Keplerlaan 1, PB 299, 2200 AG Noordwijk, The Netherlands

**Correspondence:** Carlos Toledano (toledano@goa.uva.es)

**Abstract.** Absolute calibration of Earth observation sensors is key to ensuring long term stability and interoperability, essential for long term global climate records and forecasts. The Moon provides a photometrically stable calibration source, within the range of the Earth radiometric levels and is free from atmospheric interference. However, to use this ideal calibration source one must model the variation of its disk integrated irradiance resulting from changes in Sun-Earth-Moon geometries. LIME, the Lunar Irradiance Model of the European Space Agency, is a new lunar irradiance model developed from ground-based observations acquired using a lunar photometer operating from the Izaña Atmospheric Observatory and Teide Peak, located in Tenerife, Spain. Nightly top-of-atmosphere irradiance is determined using the Langley plot method and each observation is traceable to the international system of units (SI), through the photometer calibration performed at the National Physical Laboratory. Approximately 590 lunar observations acquired between March 2018 and December 2022 currently contribute to the model parameter derivation, which builds on the widely-used ROLO (Robotic Lunar Observatory) model analytical formulation. This paper presents the strategy used to derive LIME model parameters: the characterisation of the lunar photometer, the derivation of nightly top of atmosphere lunar irradiance and a description of the model parameter derivation, along with the associated metrologically-rigorous uncertainty. The model output has been compared to PROBA-V, Pleiades, Sentinel 3B as well as to the VITO implementation of the ROLO model. Initial results indicate that LIME predicts 3% - 5% higher disk integrated lunar irradiance than the ROLO model for the visible and near-infrared channels. The model output has an expanded ($k = 2$) radiometric uncertainty of $\sim$2% at the lunar photometer wavelengths, and it is expected that planned observations until at least 2024 further constrain the model parameters in subsequent updates.

**Keywords:** Lunar; Cal/Val; calibration; ROLO; CIMEL; uncertainty; metrology; CEOS;



## 1 Introduction

Satellite Earth observation provides essential data sets for a wide range of commercial, societal and scientific applications. At present, operational long-term sustained Earth Observation programmes, such as the Copernicus Sentinels, offer reliable environmental information services to diverse users. The data acquired through these programmes provides a unique opportunity to understand the changing dynamics of our planet which has the potential to significantly influence socio-political decisions.

However, for long term environmental and climate records, the stability and interoperability of Earth observation sensors are key.

These sensors are rigorously characterised pre-flight in laboratories, however the harsh environment of space and the hardness of launch can lead to ground-to-orbit and in-orbit degradation and mean that in-flight calibration is essential to ensure satellite accuracy, stability and interoperability. In addition to onboard calibration systems, which are also susceptible to degra-

dation in space, vicarious calibration methods are often used, including the use of instrumented field sites (e.g. the automated sites of RadCalNet (Bouvet et al., 2019) and one-off measurements of ground field campaigns (Thome et al., 1993; Thome, 2001)), the use of Pseudo-Invariant Calibration Sites (PICS) (Cosnefroy et al., 1996; Lacherade et al., 2013a; Bouvet, 2014) and the use of natural phenomena, e.g. Rayleigh scattering, deep convective clouds and Sun glint (Sterckx et al., 2013; Alhammoud et al., 2018).

One important vicarious reference source is the Moon. With no atmosphere, the surface of the Moon is extremely stable long term (Kieffer, 1997). Geostationary satellite instruments sometimes observe the Moon in the "dark space corners" of their field of view and low Earth orbit sensors can be manoeuvred to observe the Moon. Many satellites already use the Moon as a calibration source, particularly to monitor long term radiometric stability. The Moon is also used to monitor atmospheric aerosols in a diurnal cycle (Barreto et al., 2019). In this sense, lunar photometry has emerged as a suitable approach to extend

aerosol remote sensing capabilities during nocturnal period, which is critical for climate studies, especially for high latitude and polar regions (Barreto et al., 2013, 2017; Berkoff et al., 2011; González et al., 2020; Román et al., 2020).

However, to use the Moon as a radiometric reference, it is necessary to model the disk lunar irradiance variations resulting from the changing lunar phase angle and lunar libration. There are many periodic cycles that apply to the Moon, Earth and Sun geometry. The cycle with the longest period is called the Saros cycle and its duration is 223 synodic months, which is 18 years,

11 days and 8 hours. After this cycle, the Earth, Moon and Sun return to the same relative geometry. The shortest cycle is the variation in phase angle which takes about 28 days between two full Moon events.

Previous models of lunar irradiance, most notably the ROLO model (RObotic Lunar Observatory, Kieffer and Stone (2005)) have proven to be valuable tools for the monitoring of radiometric stability. However, ROLO has an expected uncertainty of 5-10% (Stone and Kieffer, 2004) and is not currently used as a reference for absolute radiometric calibration. The lunar

photometry and calibration community are actively working to improve the uncertainties in the ROLO model (Smith et al., 2012; Stone et al., 2020, among others), mostly based on empirical corrections derived from observations at high altitude in pristine conditions, as those performed by Barreto et al. (2017) and Román et al. (2020)





This paper outlines the strategy used to develop a lunar irradiance model from new ground-based measurements obtained from a high altitude location. Section 2 provides a summary of lunar calibration, and ROLO, on which LIME is based. Section 3 describes the methodology for the lunar observations and derivation of LIME. Section 4 describes the calibration of the instrument used for the lunar observations, essential for providing the SI traceability and minimising the uncertainty in the resulting model. The results of the current implementation of the model are presented along with an initial comparison to several data sets. This new model is envisaged to be used as a reliable vicarious reference for absolute radiometric calibration, not only for Earth observing sensors but also for at-ground photometry for night aerosol retrieval. Model reliability has been ensured by means of a metrologically-rigorous uncertainty analysis in addition to a comprehensive validation with satellite sensors.

## 2 Lunar Calibration Background

### 2.1 The Moon as a tool for post-launch calibration

Absolute radiometric calibration of space-borne optical instruments using on-ground measurements of terrestrial ground targets is challenging as it requires accounting for the interaction of the Sun light with the atmosphere. These difficulties can be partially mitigated by acquiring measurements taken from aircraft, high above the bulk of the optically significant part of the atmosphere. Both on-ground and airborne methods can be labour-intensive, costly and reliant on favourable weather conditions.

For the radiometric intercalibration based on comparison at top-of-atmosphere reflectance level of terrestrial calibration targets, the variability of the atmosphere and environmental surface processes are an issue. These variations create difficulties in cross-calibrating instruments and ensuring continuity, especially in cases where there are gaps between instrument lifetimes.

These problems are partially addressed by the careful selection of "pseudo-invariant calibration sites" (PICS), typically located in deserts. Initiatives like RadCalNet (Bouvet et al., 2019) further contribute to overcoming these challenges by providing continuous automated in-situ measurements.

Observations indicate that the Moon is largely photometrically stable, with estimates of the change in reflectance on the order of $10^{-8}$ per year, based on the rate of meteoric impacts and the Moon's geological age (Kieffer, 1997). Additionally, the Moon exhibits a similar reflectance (approximately 10% throughout the visible spectral domain) to Earth, making it suitable for calibrating sensors within their radiometric dynamic range. Unlike very bright targets such as clouds, the Moon does not exceed the dynamic range of the sensors. Moreover, the Moon is unaffected by atmospheric interference that is typically associated with the use of terrestrial targets, making it an ideal calibration target.

Any SI-traceable model that provides an absolute irradiance of the Moon, considering phase and libration, along with a comprehensive uncertainty analysis, could be utilized as a tool for the absolute calibration of on-orbit sensors. Moreover, since historical sensors have regularly observed the Moon, such a model could facilitate the recalibration and reanalysis of historical data. This would significantly enhance the accuracy of our historical climate records by extending the time base of reliable, SI-traceable climate data records, and reduce uncertainty in our climate forecasts.



## 2.2 Robotic Lunar Observatory (ROLO)

The first observations of the Moon to be considered fully radiometrically calibrated were those of Lane and Irvine (1973) as part of a programme at Harvard University which observed the whole disk of the Moon and several bright planets. Prior studies of the phase curves of the Moon were limited to selected regions of the Moon rather than the complete lunar disk or focused on selective wavelength ranges. Irvine and Lane's model of lunar irradiance, was a huge step forward from any previous work, covering phase angles 6°-120° and 9 narrow spectral bands between 350 nm and 1000 nm. However, their model did not account for lunar libration or oppositional effect –a sharp increase in the brightness of the Moon as phase angle approaches zero (opposition)–, thought to be resulting from shadow hiding and/or coherent backscattering (Muinonen et al., 2002). The work by Lane and Irvine (1973) was developed further by Kieffer and Stone, who produced the ROLO (Robotic Lunar Observatory) model from 8 years of images taken by two telescopes at the ROLO observatory at the US Geological Survey field centre in Flagstaff, Arizona from March 1996 to September 2003 (Stone and Kieffer, 2002). The observations covered a wide range of observable libration angles and phase angles ±90°. Observations were obtained in 23 VNIR (Visible and Near Infrared) and 9 SWIR (Short-Wave Infrared) passbands selected to allow 7 of the VNIR bands to coincide with operational Earth Observing System (EOS) instruments, and 16 being the Nyquist pairs in standard astronomical bands, used in colour corrections of different colour temperatures. Observations of selected stars were also acquired in addition to the lunar measurements to allow determination of atmospheric extinction to use in correction of the lunar acquisitions. The star Vega was used as the absolute radiometric standard to tie the lunar irradiance scale to, determined using astronomical literature and using observations from the ROLO telescopes.

This star-based calibration method resulted in lunar reflectance spectra that had band-to-band deviations which were not consistent with the measured reflectance spectra of returned Apollo lunar samples which were used in the spectral interpolation of the ROLO model. The variation in the spectral (absolute) band scaling results from the significant difference between the zenith angle for the star and the lunar zenith angle, introducing different path lengths and spectral absorption features in the measurement.

Kieffer and Stone (2005) therefore proposed a correction based on Apollo lunar rock samples in an attempt to correct the spectrum for these problems with the absolute calibration. They proposed a set of parameters that smooth the ROLO model output spectrally at one configuration: a phase angle of 7 degrees and zero degrees of libration. This smoothing process uses the spectrum obtained from lunar rocks brought back during the Apollo missions at a specific mix of (5% breccia and 95% soil) because no individual Apollo sample is representative of the entire Moon. The resulting spectrum is shown in Figure 1.

### 2.3 Developments of the ROLO model

The Global Space-Based Inter-Calibration System (GSICS) developed an implementation of the ROLO model providing an accessible tool known as GIRO (EUMETSAT, 2015).

The GIRO software takes in the spectral response function of a satellite sensor, the time of observation, and position of the satellite at that time and the sensor lunar irradiance acquisition. From the time and position, it uses the NASA NAIF



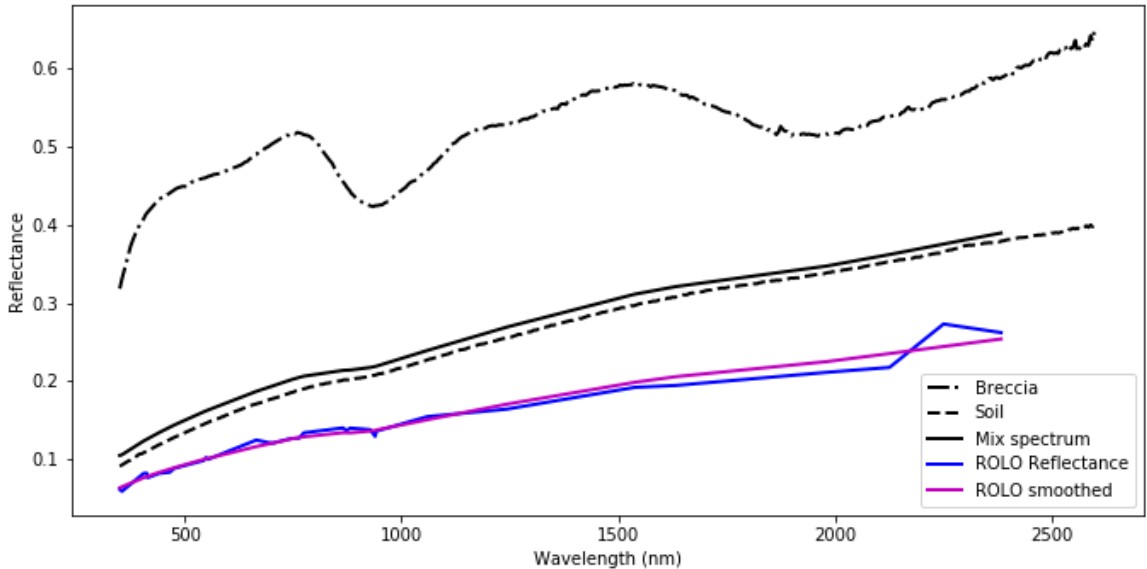

**Figure 1.** ROLO reflectance before and after smoothing process, including measured spectra from the two lunar samples (breccia and soil), as well as the composite spectrum (adapted from Kieffer and Stone, 2005)

(Navigation and Ancillary Information Facility) SPICE tool (Acton, 1996) to calculate the selenographic coordinates (lunar phase and libration angles, $g, \theta, \psi, \phi$) and solar and lunar distances and the ROLO model is used to calculate the modelled lunar reflectance (from the fit parameters).

EUMETSAT has undertaken an extensive comparison between the GIRO and the original ROLO implementation. The GIRO implementation was compared to the ROLO with perturbations of all input parameters, resulting in thousands of simulations. It was reported by Stone and Wagner (2018) that there were only differences related to numerical instabilities. Therefore, ROLO and GIRO can be considered identical.

Stone and Kieffer (2004) performed an assessment on the uncertainty involved in the ROLO lunar irradiance. They identify the ROLO atmospheric extinction correction algorithm as the most important source of error, accounting for the 5-10% of absolute accuracy uncertainty of the irradiance model (Stone and Kieffer, 2004).

Lacherade et al. (2013b) observed a phase angle dependency of the ROLO calibration up to 6% when comparing ROLO outputs and SEVIRI lunar irradiance measurements. A similar dependency with phase angle between the PLEIADES (space Earth Observation program of the French Space Agency -CNES) lunar-irradiances with the irradiances predicted by ROLO/-GIRO was observed (Colzy et al., 2017). This phase angle dependency of the GIRO was further investigated by (Barreto et al., 2016) by making Aerosol Optical Depth measurements from a high altitude observatory in Tenerife, Spain using the Cimel



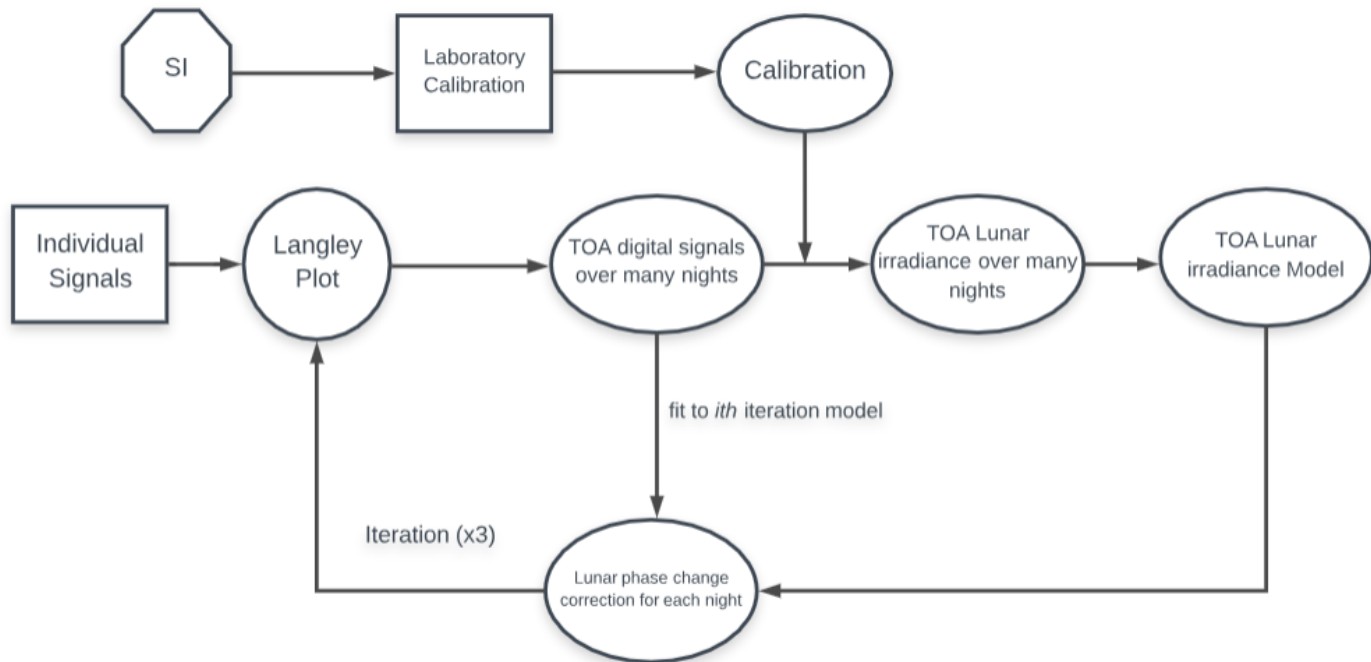

**Figure 2.** Overview of the development of the LIME model

CE318-T photometer, where they noted a dependency of the aerosol optical depth measurements on the lunar phase angle. It was inconclusive as to whether this systematic error was the result of instrumental issues or inaccuracy of the ROLO model.

Other in-orbit direct measurements of lunar irradiance have also indicated a relative difference of up to 10% with the corresponding predictions of the ROLO/GIRO model. These comparisons and applications of the current models indicate that further work is required to develop an SI-traceable absolute irradiance model of the Moon.

## 3    LIME Overview

### 3.1    Overview

The Lunar Irradiance Model of ESA (LIME) is developed from SI-traceable observations of the Moon acquired by a Cimel CE318-TP9 photometer from high altitude locations, accompanied by a rigorous uncertainty analysis from calibration, through individual measurements to the model fit. Nightly top of atmosphere (TOA) irradiance is determined using a modified, iterative Langley plot method and fit to a model based on the ROLO equations using specific coefficients for the Cimel spectral bands (see section 6.2). Figure 2 visualizes the process followed for the development of LIME.





## 3.2  Metrological Approach and Uncertainty Analysis

A key attribute of the LIME model is a rigorous uncertainty analysis and the ambitious target of a sub-2% uncertainty in the resultant model. Uncertainty contributions for each step towards the model (lunar photometer radiometric calibration, individual measurement, derivation of TOA lunar irradiance, model fit) are considered independently, taking into account the measurement function – that is, the equation that calculates the measurand from input quantities. We then use the principles of the Guide to the Expression of Uncertainties in Measurement (the GUM JCGM, 2008) to propagate uncertainties from the laboratory calibration to the TOA lunar irradiance model parameters and output. The GUM describes two methods for performing uncertainty analysis: the Law of Propagation of Uncertainties and the Monte Carlo methods (JCGM101, 2008). When propagating uncertainties from radiometric calibration to nightly TOA lunar irradiance we use the Law of Propagation of Uncertainties. When considering the uncertainty in the model parameters we use a Monte Carlo approach.

## 3.3  Instrument

The Cimel Electronique CE318-TP9 Sun-sky-Moon photometer was the chosen instrument for the lunar observations. The photometer is well known for its reliability through its use in the NASA AERONET Programme (https://aeronet.gsfc.nasa.gov/, Holben et al., 1998), the globally-distributed network of Sun photometers used for providing spatial and temporal extent of aerosol concentrations and properties, for satellite validation and assessing the influence of aerosols on climate change.

The CIMEL 318-TP9 is an upgrade to the standard AERONET model and meets many of the requirements for the low uncertainty lunar observations. It is a multi-spectral filter radiometer which is a weather-hardy and robotically-pointed to acquire measurements of Sun, Moon and sky. It has the necessary tracking capability and the dynamic range and linearity needed to cover both solar and lunar observations. It is fully automated and day and night measurements are performed following the AERONET schedule. It comes with 9 standard filters centred on 340, 380, 440, 500, 675, 870, 937, 1020, 1640 nm, and is equipped with 2 detectors in the sensor head: Silicon and InGaAs. The 1020 nm filter is used in both the Si and InGaAs detectors for quality control purposes. It also has built-in polarisation measurement capability, with 3 polarisers oriented at $0°$, $60°$ and $120°$.

The photometer acquires measurement at varying electronic gains, allowing coverage of the wide dynamic range. The electronic gains are automatically set depending on the target source, so the SUN gain is the lowest, and MOON and SKY gain the highest, with AUREOLE (for sky radiance measurement in the solar aureole) in between.

## 3.4  Location

To characterize the extra-terrestrial irradiance of the Moon, $E_0^m$, or its reflectance, $A$, it is essential to have as many high-quality Langley plots as possible, so it is necessary to carry out measurements in high-altitude stations (Shaw, 1979). For this reason the Teide Peak Observatory was been selected as the main measurement site for the derivation of lunar TOA irradiances. The Izaña Observatory is used as a backup station for the season of the year in which the Teide Peak Observatory is not in operation.



The Teide Peak Observatory (3555 m a.s.l.) is one of the highest altitude stations of AERONET and is managed by the Izaña Atmospheric Research Center (CIAI, https://izana.aemet.es/) from the Meteorological State Agency of Spain (AEMET). CIAI has its own main observatory at the Izaña Mountain (2401 m a.s.l.), 15 km from the Teide Peak Observatory. Both
stations present optimum conditions required to derive TOA irradiance by the Langley plot method: they are located in the free troposphere with very low aerosol content, water vapour column and molecular (Rayleigh) optical depth. The low latitude (28°N) reduces the time needed to acquire Sun and Moon observations at a wide air mass range (i.e. solar elevations). Izaña also experiences 243 clear sky days per year (Toledano et al., 2018), which is critical because even thin high clouds significantly perturb the Langley calibration. Regarding the aerosol climatology, dominant background conditions are expected at both sites,
with more than 69% of the time under pristine conditions (AOD<0.1 and Ångström Exponent>0.75), as shown in Barreto et al. (2022). Only Saharan mineral dust episodes (about 20% of days) affect these high-altitude locations, mainly in summer (July and August). Finally, the CIAI has permanent, experienced staff, as it has been involved in AERONET for more than 20 years, and is accessible throughout the year. The suitability of Izaña to be an absolute calibration site by means of the Langley technique has been demonstrated in Toledano et al. (2018) and Cuevas et al. (2019). Izaña is one of the two absolute calibration
sites of key photometric networks worldwide: NASA AERONET and GAW-PFR.

## 3.5   Strategy for extraterrestrial Moon irradiance retrieval

The classical Langley plot method is based in the Beer's Law (Thomason et al., 1982; WMO, 2016) which is strictly only applicable to monochromatic light, but is generally accepted for narrow spectral bands with weak gas absorption. It is commonly used in the derivation of aerosol optical depth during daylight observations, where Beer's law describes the attenuation of the
Sun's irradiance by the atmosphere.

The Langley plot method is based on the hypothesis that the properties of the atmosphere remain constant during the time needed to perform the measurements used in the linear regression, but this condition is not generally completely satisfied. To minimize the effect of changes in the atmospheric conditions, this method is better applicable if the measurements are taken at high altitude locations, where the atmospheric attenuation and variability is low. The atmospheric conditions, however, are
not always optimum, thus this method results in significant variability of the retrieved lunar extraterrestrial irradiance, and it is necessary to have a large number of measurements to statistically filter the most likely outlier lunar irradiance values.

The difference between the Langley plot method applied to the Sun and the Moon is that we need to account for the continuously changing phase and libration angles of the Moon throughout the Langley period. Therefore, even when the photometer measurements are normalised to the mean Earth-Moon and Sun-Moon distances, it is necessary to consider how the
lunar extraterrestrial irradiance (and the corresponding raw measurements extrapolated at a null atmospheric optical thickness) is dependent on time, and the values obtained each night will be different.

Equation (1) is the application of Beer's law to the photometer measurements.

$$V^m(\lambda, t) = V_0^m(\lambda)e^{-m(\theta)\tau(\lambda)} \tag{1}$$




where $V^m(\lambda, t)$ is the photometer output signal (in digital counts) when it measures pointing to the Moon, $V_0^m(\lambda)$ represents the lunar top-of-atmosphere signal of the photometer, $m$ is the airmass calculated using the equation which is a function of the Sun zenith angle $\theta$ in Kasten and Young (1989), and $\tau(\lambda)$ is the spectral total optical depth. The Langley plot method considers that the properties of the atmosphere remain constant with time, and therefore $\tau$ is written as not time dependent. Taking logarithms on both sides of the equation (1) gives,

$$ln(V^s(\lambda, t)) = ln(V_0^s(\lambda)) - m(\theta)\tau_\lambda \tag{2}$$

which represents a linear relation between $ln(V_\lambda)$ and $m(\theta)$. By making observations over a relative airmass 2-5 and fitting a straight line to the results, one can determine $ln(V_{0,\lambda})$, the logarithm of the top of atmosphere signal, as the y-intercept of the linear regression of measurements vs. airmass.

The Langley plot method allows a determination of the virtually measured top-of-atmosphere signal of the photometer by effectively correcting the effect of the atmospheric extinction on the ground measurements. In this way, it is not necessary to know a priori the gas and aerosol extinction, which is general unknown. Thus, for each suitable night, extraterrestrial lunar irradiance for a specific phase angle is obtained.

To account for the change in lunar irradiance due to minute changes in phase angle during the Langley period, the typical Langley plot method is modified to include an iterative step. Figure 3 outlines this iterative process. In the first iteration, it is considered that the top of atmosphere irradiance of the Moon remains constant during the Langley measurements. Applying this approach to the complete set of night data (several years of measurements) a first estimation of the lunar reflectance, $A$, is obtained. These first reflectance values are used to adjust a lunar reflectance model, based on the ROLO equations, in order to have an estimation of the lunar irradiance change during the Langley measurements in the next iteration. This first estimate of the lunar reflectance model is used to perform a new set of nocturnal Langley plots, this time taking into account the phase change of the Moon irradiance along the Langley duration, using the correction:

$$V'(\lambda, t) = V(\lambda, t)\frac{A(t_{ref}, \lambda)}{A(t, \lambda)} \tag{3}$$

where, $V$ and $V'$ are the photometer signal before and after phase change correction, $t_{ref}$ is the mean time of each Langley plot, thus the time corresponding to $V_0^m$ from each Langley fit. The iterative process converges rapidly, only 3 iterations are required. Finally a corrected set of lunar TOA irradiances and phase angles are obtained.

## 4 Calibration

Many current post launch satellite calibration systems are not traceable to SI (Bouvet et al., 2019). In the development of LIME this issue is addressed by deriving the model from SI traceable observations through the photometer irradiance calibration, performed at NPL.





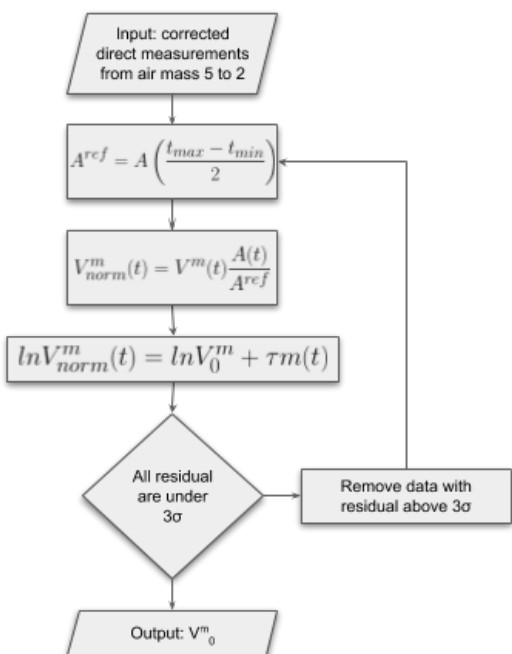

**Figure 3.** Iterative correction scheme of TOA irradiance over a Langley period (single night Moon observations).

A detailed characterisation of the lunar photometer was performed and accompanied by a rigorous uncertainty analysis following metrological principles. The characterisation included an assessment of linearity across the wide dynamic range of
signal expected across the lunar cycle; thermal sensitivity characterisation of the instrument to determine corrections to apply to data obtained at a wide range of ambient temperatures; and spectral irradiance calibration coefficients for each spectral channel of the instrument.

## 4.1 Linearity

For the Cimel photometer to measure the signal range over the lunar phase and librations, the instrument has a wide dynamic
range. It was necessary to perform tests to ensure it has a linear response over this range. These tests were performed at the NPL linearity faciliy using best practice for linearity characterisation, the double aperture method (Theocharous, 2012). This is based on the superposition principle.





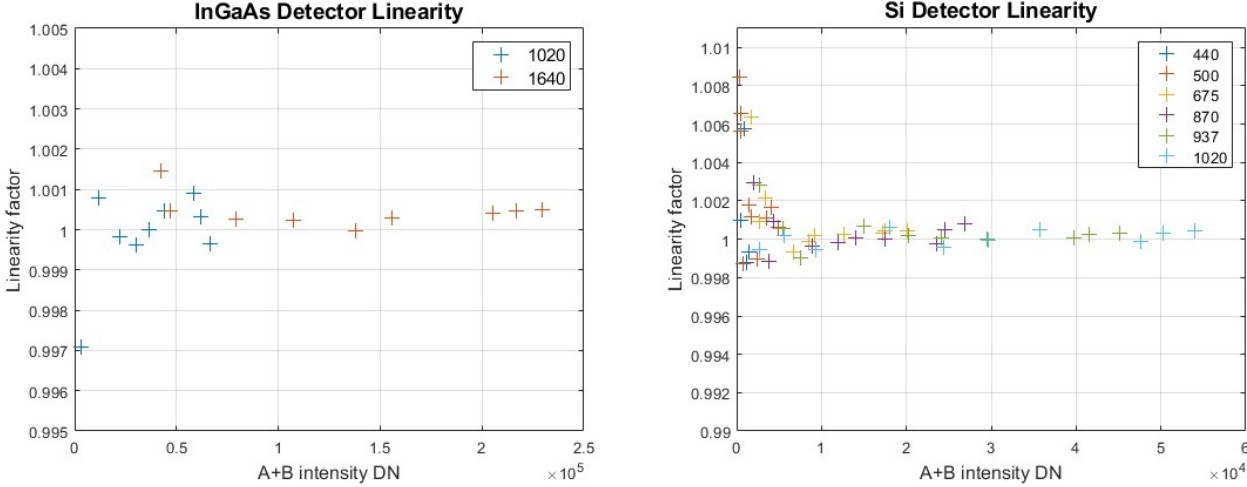

**Figure 4.** Linearity results for the InGaAs detector (left) and for the Silicon detector (right) of the Cimel radiometer.

The NPL facility used a double aperture linearity wheel, and neutral density filters to vary the levels of radiation. The facility is fully automated, and allows the testing of the linear response of a detector over the spectral range 200 nm to 20 $\mu$m 
(depending on the light source used).

The double aperture wheel is set into four different positions during the measurements, where for position A and B respectively bottom or top half on the illumination beam is baffled, A+B position gives the full light reading and D is external dark reading. A tungsten strip lamp is used as a source. Measurements were repeated a number of times and each sequence consisted of dark reading, A reading, B reading, A+B reading and then B reading, A reading and dark reading in the end. For a linear 
response we would expect the signal when both the apertures are open to equal the sum of the signal through each aperture individually. A linearity factor, $L(V_{A+B})$, is then calculated using equation (4).

$$L(V_{A+B}) = \frac{V_{A+B}}{(V_A + V_B)} \tag{4}$$

A linearity factor was calculated for illumination levels varied by changing neutral density filters, then the results were averaged for the final estimation of linearity for each Cimel spectral channel. For wavelengths of 500 nm and beyond, the non-
260 linearity was observed to be less than 0.1 % with a standard deviation of values for different levels of less than 0.1 % and the results show no systematic pattern (Figure 4). Therefore, here non-linearity was considered negligible. At shorter wavelengths any non-linearity was indistinguishable from the noise.

## 4.2 Temperature sensitivity

The temperatures at the observation sites varies significantly, between the two sites as well as seasonally (Table 1). As the 
265 instrument is not thermally stabilised, the thermal sensitivity of the instrument was assessed to determine and temperature





correction to compensate for the effect of the temperature variation during the realisation of day and night Langley plots. This is particularly important for 1020 nm, where the Silicon detector is particularly temperature sensitive, but is also necessary to determine for all spectral channels.

| Izaña | Jan | Feb | Mar | Apr | May | Jun | Jul | Aug | Sep | Oct | Nov | Dec |
|---|---|---|---|---|---|---|---|---|---|---|---|---|
| Temp. Max. [ºC] | 7.2 | 8.2 | 9.3 | 11.1 | 14.1 | 18.4 | 22.5 | 22.4 | 18.2 | 13.9 | 10.7 | 7.1 |
| Temp. Min. [ºC] | 0.8 | 1.4 | 2.0 | 2.9 | 5.4 | 9.4 | 13.5 | 13.5 | 10.1 | 6.7 | 4.2 | 1.9 |

| Teide Peak | Jan | Feb | Mar | Apr | May | Jun | Jul | Aug | Sep | Oct | Nov | Dec |
|---|---|---|---|---|---|---|---|---|---|---|---|---|
| Temp. Max. [ºC] | 7.7 | 8.0 | 8.7 | 11.7 | 14.1 | 16.6 | 18.7 | 18.8 | 15.1 | 13.0 | 9.9 | 7.5 |
| Temp. Min. [ºC] | -9.5 | -12.0 | -8.1 | -5.7 | -2.7 | 1.8 | 5.0 | 5.0 | 1.9 | -3.5 | -5.9 | -8.2 |

**Table 1.** Monthly maximum and minimum daily mean temperatures for the Izaña station (upper table) in the period 1971-2000; and for Teide Peak (lower table), in this case preliminary statistics from 2013-2016.

The temperature sensitivity was characterised at the University of Valladolid in a CLIMATS-TM thermal chamber. This consists of a stainless-steel cabinet with a gridding where the photometer can be positioned, and the right side has an 80 mm diameter aperture. The photometer is aligned with an integrating sphere (Labsphere 10" diameter), illuminated with a 100 W lamp, and powered by a stabilized power supply (Agilent E3634A). Two independent tests were carried out where the lunar photometer sensor head performs measurements on the MOON gain scenarios at temperatures ranging from $+50$°C to $-40$°C during a period of 4 hours, which is a rate of change of about 0.3 °C/min.

The irradiance, $E_T$ is measured in the temperature chamber at different temperatures $T$[°C], while the detector is illuminated with a stable light source. The acquired data are then fitted to the following model:

$$E_T = E_c + c_1(T - T_{ref}) + c_2(T - T_{ref})^2 \tag{5}$$

where $T_{ref}$ is the reference temperature (25°C) and $E_c$ is the irradiance of the sources measured at the reference temperature. The measurements for all spectral channels in the range 440-1640 nm are shown in Figure 5.

The coefficients $c_1$ and $c_2$ are then used in a temperature correction factor (equation (6)) which is applied to all lunar measurements, correcting for the differing temperatures during observation. The temperature corrections $F_{T,i}$ for the spectral channel $i$ were applied to the raw data used to determine the spectral irradiance and radiance calibration coefficients at NPL.

$$F_T = [1 + c_{1,i}(T - T_{ref}) + c_{2,i}(T - T_{ref})^2] \tag{6}$$

Uncertainties associated with the temperature sensitivity coefficients derived from a measurement sequence are very small.Therefore we repeated the measurements in the thermal chamber and compared the two sets of temperature sensitivity coefficients. The





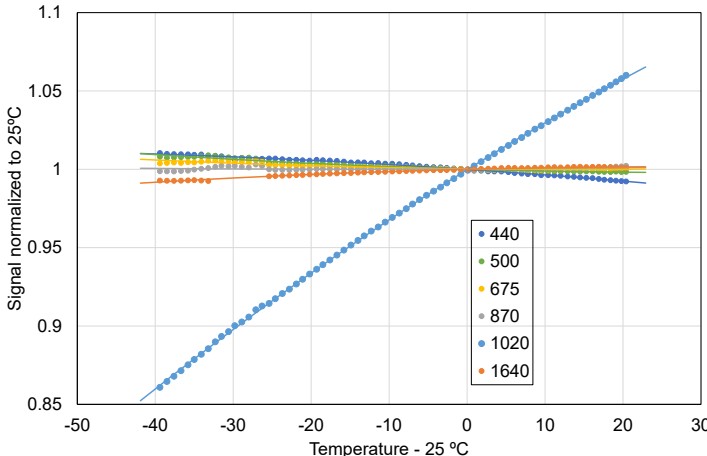

**Figure 5.** Observed temperature sensitivity plot for the lunar photometer channels (440 to 1640 nm), where the signals have been normalized to the value at 25ºC.

| Cimel channel [nm] | Uncertainty[%] |
|---|---|
| 1020 | 0.13 |
| 1640 | 0.003 |
| 870 | 0.18 |
| 675 | 0.17 |
| 440 | 0.05 |
| 500 | 0.15 |

**Table 2.** Estimates of the relative uncertainty associated with the temperature correction for irradiance measurements in each lunar photometer spectral channel.

difference between the temperature corrections associated with each set of coefficients was calculated. This difference is temperature dependent and is zero for the reference temperature (25°C). In order to provide example uncertainties, the difference between the correction calculated with each set of coefficients was determined at 11.3 °C as that is the mean temperature during Langley plots at Izaña observatory from June 2014 to October 2017. The values are given in Table 2. We consider

this is a better estimation of the uncertainty in the temperature correction, as it involves the repeatability of the entire thermal characterization.



### 4.3 Spectral Irradiance Responsivity

The irradiance responsivity of each lunar photometer spectral channel was assessed at NPL by measuring the response of each of the detectors in turn with two different sources of known irradiance at a variety of distances, to ensure the wide dynamic range required for the lunar observations was covered.

Irradiance sources calibrated at NPL are traceable to the primary standard, the cryogenic radiometer where electrical power is compared to optical power in a cryogenic blackbody cavity. The calibration is then transferred via a laser to a trap detector (an arrangement of Si photodiodes that reduce signal loss by reflectance to negligible levels). This in turn is used to calibrate a filter radiometer, which measures the radiance of a 3000 K blackbody source in the spectral band of the filter. From this, and Planck's Law, the temperature of the blackbody is known accurately, and hence the radiance at other temperatures. With an appropriate geometric system, consisting of two apertures, the radiance of the blackbody can be compared directly with the irradiance of a FEL lamp at 500 mm from the reference frame, using a monochromator to measure each wavelength. The FEL lamp is therefore a reference source of known spectral irradiance, and was used in the Cimel calibration as the calibrated reference source. To achieve lunar irradiance levels, a transfer radiometer was used to step down to a lower power source.

The calibration coefficient for each spectral band was determined by equation (7):

$$C_{\overline{E},Cimel}(\lambda_i) = \frac{(\sum_j E_{lamp,x}(\lambda_j)\xi_j(\lambda_j)\delta\lambda)F_T}{G_{ratio}[D_{Cimel,lamp,x}(\lambda_i) - D_{Cimel.dark}(\lambda_i)]} + 0 \tag{7}$$

Where $C_{\overline{E},Cimel}(\lambda_i)$ is the band-integrated irradiance calibration coefficient for band $i$ of the lunar photometer at wavelength $\lambda$;

$E_{lamp,x}(\lambda_j)\xi_j(\lambda_j)$ is the irradiance of the lamp at wavelength $\lambda$ and distance $x$;

$\xi_j(\lambda_j)$ is the normalised (for unit area) spectral response function at wavelength $\lambda_j$ defined at equispaced wavelengths separated by $\delta\lambda$;

$F_T$ is the temperature correction from the calibration instrument temperature to the nominal reference temperature of 25 °C;

$G_{ratio}$ is the gain ratio from the gain of measurement (e.g. SUN or AUR) to the MOON gain. Depending on the measurement target, the photometer switches the electronic gain automatically. The linearity was characterized for all gain settings (section 4.1). The nominal values are:

$$\frac{C_{SUN}}{C_{MOON}} = 4096; \qquad \frac{C_{AUR}}{C_{MOON}} = 32 \tag{8}$$

$D_{Cimel,lamp,x}(\lambda_i)$ s the Cimel signal for channel at wavelength $\lambda$ when looking at the lamp at distance $x$;

$D_{Cimel.dark}(\lambda_i)$ Is the dark signal for channel at wavelength $\lambda$;

0 represents the approximations in the form of the equation, in particular that the Cimel is linear and that the summation on the numerator is an appropriate approximation for the spectral integral.





## 4.4  Calibration uncertainty Analysis

As described in section 3.2, rigorous uncertainty analysis was a key attribute of the LIME model where the target is to reach an uncertainty <2% in the resultant model. By calibrating the lunar photometer, traceable to the primary standard of the cryogenic radiometer at NPL, low uncertainty in the photometer calibration was achievable. The uncertainty analysis for the radiometric
calibration followed the principles in the GUM (JCGM, 2008), using the Law of Propagation of Uncertainty.

The Law of Propagation of Uncertainties applies a locally-linear approximation to the measurement function $f$ and propagates standard uncertainties (that is the standard deviation of the probability distribution from which the unknown measurement error is drawn) through this approximation. It can be written as a summation as:

$$u_c^2(y) = \sum_{i=1}^{n} \left(\frac{\partial f}{\partial x_i}\right)^2 u^2(x_i) + 2 \sum_{i=1}^{n-1} \sum_{j=i+1}^{n} \frac{\partial f}{\partial x_i} \frac{\partial f}{\partial x_j} u(x_i, x_j) \tag{9}$$

where $u_c^2(y)$ represents the combined variance, the first term adds the uncertainty contributions from the input quantities in quadrature, and the second term deals with error correlation (common errors) between the different input quantities. The partial derivatives $\left(\frac{\partial f}{\partial x_i}\right)$ are the sensitivity coefficients, which convert an uncertainty associated with the input quantity $x_i$ into an uncertainty associated with the measurand. The term $u(x_i, x_j)$ is the covariance associated with pairs of input quantities.

It was important to distinguish "common uncertainties" from "common errors". Two measurements at different distances may have a common uncertainty associated with noise, but they will have different errors. On the other hand, the uncertainty associated with lamp calibration will have a common error at the two different distances. By considering what is common and what changes from one measurement to another, we can get a meaningful uncertainty associated with an average of those measurements: the uncertainties associated with effects that have common errors will not reduce on averaging, those associated with effects whose error changes, will reduce on averaging.

The irradiance responsivity was measured at NPL using several methods in order to cover the range of signal levels appropriate for the lunar observations as described in section 4.3. The band-integrated irradiance calibration coefficient is determined for each method at varied lamp-photometer distance along with an associated uncertainty.

We consider for each method the measurement equation, the general form applicable to each method given in equation (7). The uncertainty for each term in the measurement equation and its magnitude was determined and then categorised by the
following error correlation structures.

a) Fully independent uncertainties corresponding to errors (e.g. noise) which vary from observation to observation (i.e. is different at different distances and for the different methods).

b) Fully common uncertainties (e.g. SRF) where the error is (almost) identical for all measurements with all sources at all distances.

c) Lamp uncertainties (e.g. its calibration) where the error is common to all methods that use the same lamp.





d) Method common uncertainties where the error is common to the measurements at different distances with this method, but which is different for other methods.

All the considered uncertainty sources in the calibration of the lunar photometer are provided in table 3. The final calibration coefficients and their associated standard ($k = 1$) and expanded ($k = 2$) uncertainties are given in table 4.

| Effect | | Noise | Lamp irradiance uncertainty | Alignment | Current stability | Lamp filament offset |
|---|---|---|---|---|---|---|
| Error correlation | Category | a | c | c | a | c |
| Uncertainty Magnitude | 440 nm | 0.44% | 0.52% | 0.20% | 0.13% | 0.10% |
| | 500 nm | 0.10% | 0.47% | 0.20% | 0.12% | 0.10% |
| | 675 nm | 0.05% | 0.40% | 0.20% | 0.09% | 0.10% |
| | 870 nm | 0.04% | 0.35% | 0.20% | 0.07% | 0.10% |
| | 1020 nm (Silicon) | 0.06% | 0.43% | 0.20% | 0.06% | 0.10% |
| | 1020 nm (InGaAs) | 0.04% | 0.43% | 0.20% | 0.06% | 0.10% |
| | 1640 nm | 0.05% | 0.44% | 0.20% | 0.04% | 0.10% |

| | Distance settings | Cimel detector offset | Lamp aging | Spectral Interpolation of FEL spectrum | Cimel Spectral Response Function | Assumptions in the form of the equations (+0) |
|---|---|---|---|---|---|---|
| Error correlation | d | b | c | c | b | b |
| Uncertainty Magnitude | 0.10% | 0.81% | 0.30% | 0.17% | Negligible | Negligible |
| | 0.10% | 0.81% | 0.30% | 0.02% | Negligible | Negligible |
| | 0.10% | 0.81% | 0.30% | 0.05% | Negligible | Negligible |
| | 0.10% | 0.81% | 0.30% | 0.02% | Negligible | Negligible |
| | 0.10% | 0.81% | 0.30% | 0.03% | Negligible | Negligible |
| | 0.10% | 0.81% | 0.30% | 0.03% | Negligible | Negligible |
| | 0.10% | 0.81% | 0.30% | 0.05 | Negligible | Negligible |

**Table 3.** Uncertainties associated with the calibration of the lunar photometer. See text for the description of categories 'a', 'b', 'c', 'd' of error correlation structures.



| Spectral Channel | MOON calibration Coefficient | Standard uncertainty $(k=1)$ | Expanded uncertainty $(k=2)$ |
|---|---|---|---|
| 440 nm Si | $5.759 \times 10^{-10}$ | 0.97% | 1.94% |
| 500 nm Si | $4.481 \times 10^{-10}$ | 0.96% | 1.91% |
| 675 nm Si | $3.205 \times 10^{-10}$ | 0.92% | 1.85% |
| 870 nm Si | $2.547 \times 10^{-10}$ | 0.91% | 1.82% |
| 1020 nm Si | $2.735 \times 10^{-10}$ | 1.05% | 2.11% |
| 1020 nm InGaAs | $2.119 \times 10^{-10}$ | 1.01% | 2.03% |
| 1640 nm InGaAs | $4.893 \times 10^{-11}$ | 1.06% | 2.11% |

**Table 4.** Calibration coefficients for each Cimel photometer spectral band used in LIME, and associated uncertainty.

## 5   Derivation of nightly TOA Lunar irradiance and associated uncertainty

Top-of-atmosphere lunar irradiance is determined each night through lunar phase angles spaning from -90º to +90º by acquiring a large number of measurements over relative airmass 2-5 (lunar zenith angles between 60º and 78º) and determining TOA signal using the Langley plot method. Observations were made at the Izaña Atmospheric observatory in the colder winter months and at the Teide Peak in the summer.

### 5.1   Measurement Acquisition Details and Data Processing

The ESA lunar photometer (Cimel with serial number #1088) has been installed and operational since March 2018. In the current iteration of LIME, the instrument has provided approximately 590 nights of lunar acquisitions suitable for Langley plot over more than 5 years of measurements. On 60% of the days the photometer has operated at Izaña, whereas on 40% of the days it has operated at Teide Peak.

As well as the ESA lunar photometer, another instrument ('master' instrument serial #933) has been acquiring lunar observations over a period of 9 years, providing more than 1100 acquisitions. The concurrent operation of this and other reference ('master') photometers at the Izaña site provides an opportunity for monitoring the stability of the ESA photometer between laboratory calibrations. A careful comparison between both instruments was carried out in the framework of the LIME project (https://calvalportal.ceos.org/lime-documents). These comparisons, together with the analysis of solar Langley plots, reveals a very small instrument drift, ranging from -0.1% for the SWIR channels up to -0.8% for the 440 nm channel, over more than 4 years of operation.

The instrument direct Moon observations consists of groups of 3 acquisitions (dark current is automatically subtracted) within 1 minute interval, for all spectral channels. The CAELIS software tool (Fuertes et al., 2018) is used to automatically digest those data and produce the Langley plot (see section 3.5) plus some statistical indicators of the fit quality. Moon zenith angles as well as Earth-Moon-Sun distances are obtained from the SPICE astronomic library. CAELIS also produces real time





data flags according to the meta-data provided by the Cimel instrument, as well as aerosol optical depth and precipitable water vapor (González et al., 2020). All this information is used to monitor the instrument performance on a daily basis.

## 5.2 Deriving Lunar irradiance from observations

SI-traceable Lunar extraterrestrial irradiance, $E_0^m(\lambda, t)$, was determined on each night of observation multiplying lunar top-of-atmosphere signals of the photometer, $V_0^m(\lambda, t)$, obtained by the nocturnal Langley plots, by the absolute radiometric calibration coefficient, $C_{\bar{E}, Cimel}(\lambda_i)$ determined in the characterization carried out in the NPL facilities. as described in section 4.3:

$$E_0^m(\lambda, t) = V_0^m(\lambda, t) C_{\bar{E}, Cimel}(\lambda_i) \tag{10}$$

Additionally, the lunar disk-equivalent albedo, $A$, or more briefly the lunar reflectance, was also determined. $A$, is defined 385 as follows:

$$A(\lambda, t) = \frac{E_0^m(\lambda, t) \cdot \pi}{\Omega_m E_0^s(\lambda)} \tag{11}$$

where $E_0^m$ is the lunar extraterrestrial irradiance, $E_0^s$ is the solar extraterrestrial irradiance and $\Omega_m$ is the solid angle of the Moon at mean distance (384400 km), which takes a value of $6.4177\mathrm{e}{-5}$ sr. The TSIS-1 solar spectrum (Coddington et al., 2021) is used for this purpose.

## 390 5.3 TOA lunar irradiance measurement uncertainty analysis

The uncertainty analysis for the individual measurements contributing to the Langley plots, and subsequent derivation of the nightly TOA irradiance, follows the Law of propagation of uncertainty, as described in section 4.4. In this analysis we only consider the final iteration of the Langley plots.

In the determination of the lunar irradiance, error correlation is important in two places. First, when one makes a Langley 395 plot and fits a straight line through data points made from individual observations of the Moon over a single night. While each measurement point will have an individual noise error (noise errors vary on timescales faster than the measurement time), other errors may be common from one measurement to another. For example, instrument calibration errors will apply to all measured values, and slowly-varying atmospheric conditions can create a measurement error that is correlated for measurement points close together in time. These correlations must be considered to obtain the right uncertainty associated with the model.

Following the same principles outlined in the uncertainty analysis for the calibration, we begin by considering the measurement equations.

### 5.3.1 Individual observations in situ

An individual measurement that goes into the Langley plot consists of a pair of air mass, $m(\Theta)$, and count signal $D'(\lambda, t)$.





$$D'(\lambda,t) = \frac{D(\lambda,t)}{F_T(\lambda)} \frac{A(t_{ref},\lambda)}{A(t,\lambda)} K_{dist} + 0 \qquad (12)$$

where $F_T(\lambda)$ is the temperature correction factor described in section 4.2.

$K_{dist}$ is a correction for the actual Sun-Moon and Earth-Moon distances relative to the standard distances:

$$K_{dist} = \left(\frac{x_{sun-moon}}{1au}\right)^2 \left(\frac{x_{earth-moon}}{384000km}\right)^2 \qquad (13)$$

and $A(t_{ref},\lambda)$ and $A(t,\lambda)$ are used to correct for the lunar phase change during the Langley period as described in section
3.5. The term '+0' represents the assumptions built into the form of the equations. For a single measurement this includes the
assumption of instrument linearity.

The airmass $m(\theta)$, equation (14), is calculated using equations in (Kasten and Young, 1989) . It is a function of lunar zenith
angle and takes a slightly different form for ozone and nitrous oxide to that for aerosols. The combined airmass is used here.

$$m = 1/(cos\theta + 0.50572(1.46468 - \theta)^{-1.6364}) \qquad (14)$$

where $\theta$ is expressed in radians.

The relative airmass range is restricted to 2-5 thereby avoiding errors in optical airmass determination that increases signif-
icantly at larger zenith angles (Russell et al., 1993). For uncertainty purposes the uncertainty associated with the airmass is
considered negligible.

The uncertainty associated with the temperature correction is outlined in section 4.2.

The correction applied for change of Moon phase and libration angles during the Langley plots, normalises the Moon direct
measurements using the ratio of the Moon reflectance at a specific time, $A(t)$, by the Moon reflectance at the Langley mean
time $A^{ref}$ . In order to obtain this parameter a first approximation of a Lunar reflectance model is used, as described in section
3.5, so this is done in an iterative process. In the first iteration $A(t)/A^{ref}$ is considered equal to 1 and a parametric model is
fitted to the retrieved Moon reflectance. The model obtained in one iteration is used in the following iteration. The uncertainty
associated with the final $A(t)/A^{ref}$ ratio is evaluated as the difference between the ratios of the last and the penultimate
iteration. Using the measurements taken at Izaña observatory from June 2014 to October 2017, the mean $A(t)/A^{ref}$ ratio for
three iterations was calculated (table 5). The difference between third and second iterations is as low as $6 \cdot 10^{-5}$, equivalent to
an uncertainty of 0.006%.

Because of the complexity of processing data within the SPICE system used in the distance correction, the Jet Propul-
sion Laboratory (JPL) does not provide numerical mechanisms for managing uncertainty information. In this correction we
have used as reference the high-accuracy lunar orientation data (MOON_ME or Moon Mean Earth/Rotation axis frame). The
only reference in the SPICE system about uncertainty in distances is related to the standard low-accuracy reference model
(IAU_MOON), which is expected to have an associated error of  0.0051° or  155 m on a great circle in the worst case, and



| Iteration | Mean $A(t)/A^{ref}$ |
|-----------|---------------------|
| **1st**   | 1                   |
| **2nd**   | 1.000535            |
| **3rd**   | 1.0000596           |

**Table 5.** Mean $A(t)/A^{ref}$ for three iterations using the measurements obtained at Izaña observatory from June 2014 to October 2017.

0.0025 degrees, or 76 m in average. Even if these values are considered (being rather conservative), we consider this a negligible error (NAIF, 2018).

Each source of uncertainty contributing to individual observations, along with the error correlation structures are summarised in Table 6; percentage uncertainty arising from each source of uncertainty at each photometer band is presented in Table 7.

### 5.3.2   Uncertainty associated with the Langley plots

We also consider the uncertainty associated with the least squares fit in the Langley plot, where we take into account the uncertainty associated with each data point and determine an uncertainty associated with the y-intercept, $ln(V_{0,\lambda})$, which is the

logarithm of the TOA signal.

Each data point was given the same relative uncertainty, taken from the standard deviation of the triplets (three individual acquisitions within 1 minute), $D(\lambda,t)$, see Table 7. A fit routine calculated the uncertainty associated with the y-intercept of the linear fit, and the $\chi^2$ of the fit. Where the observed $\chi^2$ was smaller than the expected $\chi^2$, the intercept uncertainty was accepted as given. Where the observed $\chi^2$ was larger than the expected $\chi^2$, relative uncertainty on each data point was increased by

small increments until the $\chi^2$ test was passed.

When performing the fit and applying the $\chi^2$ test, very few of the Langley plots passed using the original uncertainty on the input parameters. This indicated a potential under estimation of the uncertainty associated with the Langley plots. This could likely be the results of small changes in aerosol (and other atmospheric) properties during the Langley period. It is also reasonable to assume that using a standard deviation of the triplets, which shows instrument stability over a very short

period, underestimates the uncertainty associated with the stability of the instrument for the duration of the Langley. For those that failed the $\chi^2$ test, uncertainty on input parameters was increased incrementally until the test was passed and uncertainty associated with the y-intercept, $ln(V_0)$ was determined. A small selection required a small increase in uncertainty for each data point, and a few had high uncertainty indicating that the fit should be considered for removal from the data set or have very low weighting in the final model.

Using the values of $u(ln[V_0])$ and the $u_{new}(ln[V_0])$ where appropriate, a 'typical' uncertainty associated with the y-intercept of the Langley plots was determined and used in the Monte Carlo uncertainty analysis input parameters for the model fit uncertainty analysis (see section 6.4). The values are provided in table 8. They are in the range 0.1-0.2%, lower for longer wavelengths except for the 1640 nm channel.



| Term | Source of uncertainty | How this can be estimated | Error correlation structures (spectral and temporal dimension) |
|---|---|---|---|
| $D(\lambda,t)$ | Noise | From statistics on the triplet. Note that a standard deviation of just 3 measurements is not fully reliable, and therefore typical values should be obtained, averaging across similar scenarios. | Fully random from observation to observation and between wavelengths |
| $F_T(\lambda)$ | Uncertainty associated with coefficients $C_{1,\lambda_i}, C_{2,\lambda_i}$ | From the temperature tests of the instrument – uncertainty associated with these calculated from the difference between two corrections. | The same coefficients are used for all corrections at all times (not just during a Langley but also from night to night). Therefore although there may be a different specific error from one observation to another, as the error itself depends on temperature, because this is predictable, the errors are considered fully correlated across time. Spectrally independent (each wavelength treated separately) |
| $F_T(\lambda)$ | Uncertainty associated with instantaneous temperature $T$ | Assumed negligible | N/A |
| $K_{dist}$ | Distances | Assumed negligible | N/A |
| $\frac{A(t_{ref},\lambda)}{A(t,\lambda)}$ | Model correction uncertainty during Langley | From the difference between the penultimate and last iterations | Assumed fully correlated across all observations and all nights |
| +0 | Assumption of instrument linearity (assumptions of aerosol stability are considered below) | Assumed negligible from linearity tests | N/A |

**Table 6.** Summary of sources of uncertainty for individual lunar observations, i.e. raw signals per photometer spectral band.





| Term | Uncertainty [%] | | | | | |
|---|---|---|---|---|---|---|
| | 1640 nm | 1020 nm | 870 nm | 675 nm | 500 nm | 440 nm |
| $D$ | 0.07 | 0.05 | 0.02 | 0.01 | 0.03 | 0.04 |
| $F_T(c1, c2)$ | 0.0027 | 0.13 | 0.18 | 0.17 | 0.15 | 0.053 |
| $F_T(T)$ | 0.002 | 0.037 | 0.001 | 0.002 | 0.003 | 0.003 |
| $K_{dist}$ | 0 | 0 | 0 | 0 | 0 | 0 |
| $A_t$ | 0.006 | 0.006 | 0.006 | 0.006 | 0.006 | 0.006 |
| +0 (aerosol's diurnal cycle) | 0 | 0 | 0 | 0 | 0 | 0 |

**Table 7.** Percentage uncertainty arising from each source of raw signal uncertainty (see table 6) at each Cimel photometer band.

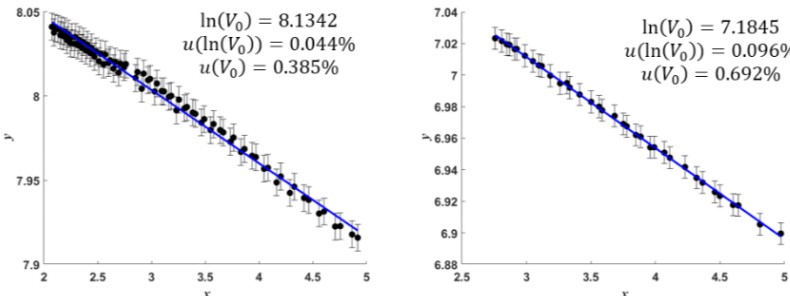

**Figure 6.** Examples of Langley plots which pass the $\chi^2$ test.

# 6  The LIME model

## 6.1  Model Concept


The model is derived from the lunar irradiance measurements from the Cimel photometer. It based on a slightly modified version of the USGS ROLO lunar model described in section 2.1. The modification in LIME is that, for each spectral band in the model, an independent set of c-coefficients has been defined, while in the original model, the c-coefficients are identical for all bands. Then the lunar reflectance $A$ for each photometer spectral band $k$ is modeled as follows:

| | 440 nm | 500 nm | 675 nm | 870 nm | 1020 nm | 1640 nm |
|---|---|---|---|---|---|---|
| $u(ln[V_0])$ | 0.21% | 0.16% | 0.13% | 0.12% | 0.12% | 0.21% |

**Table 8.** Estimated uncertainty in the y-intercept of the Langley plots for each Cimel photometer spectral band.





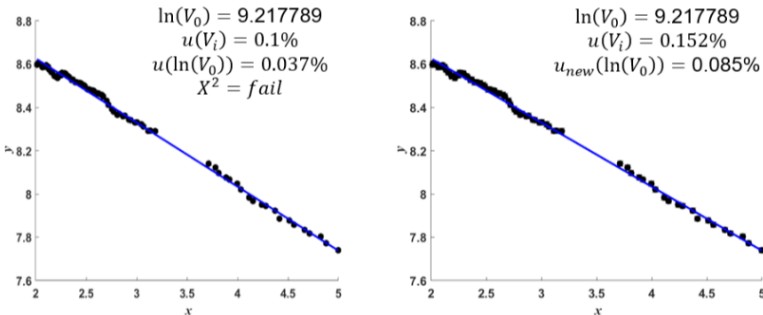

**Figure 7.** Langley before (left) and after (right) increase in uncertainty in order to pass the $\chi^2$ test.

$$ln(A_k) = \sum_{i=0}^{3} a_{ik} g^i + \sum_{i=1}^{3} b_{ik} \Phi^{2i-1} + c_{1k}\theta + c_{2k}\phi + c_{3k}\Phi\theta + c_{4k}\Phi\theta + d_{1k}e^{\frac{-g}{P_1}} + d_{2k}e^{\frac{-g}{P_2}} + d_{3k}cos(\frac{g-p3}{p4}) \qquad (15)$$

Where $ln(A)$ the natural logarithm of $A$, $g$ is the absolute phase angle [radians], $\theta$ selenographic latitude observer [degrees], $\phi$ selenographic longitude of the observer [degrees], and $\Phi$ is the selenographic longitude of the Sun [radians].

The reflectance model can be split-up in four different sections. The basic photometric function is represented by the first polynomial depending solely on the phase angle. It is a wavelength-dependent third-degree polynomial, described with the $a_k^i$

coefficients. The variations of the reflectance of the Moon due to changes in the actual area of the Moon illuminated by the Sun and driven by changes in the distribution of maria and highlands, is expressed in the second polynomial. This polynomial is depending only on the solar selenographic longitude $\Phi$. Fourth order coefficients $b_{ik}$ are defined for every wavelength. The third section, with four wavelength dependent coefficients $c_{ik}$, represents the visible part of the Moon and how it is illuminated (topographic libration). The last part of the equation is a set of parameterized exponential and cosine functions modulated by

a set of $d_{ik}$ coefficient: it is an empirical iterative least squares fitting of non-linear residuals in the irradiance, with respect to the phase angle.

### 6.2 Initial model fit at the photometer spectral bands

The strategy designed to calculate the model coefficients is divided into different steps (Figure 8), taking the lunar measurements restricted to the phase angle interval [-90;90] degrees and the geometric calculation as a starting point.

A least-squares fit is used to derive those coefficients belonging to the linear part of the model, i. e., $a$, $b$ and $c$ band specific coefficients in equation (15). All $d-$ parameters are set to zero in this first approximation. A subsequent regression is performed on the non-linear part of the equation, using the Levenberg-Marquardt method, taking into account that this non-linear part of the lunar reflectance model depends on the measurement phase angle. Therefore $d$ and $p$ parameters are calculated from the residuals calculated with previous steps. For convenience in the first iteration, all $a$ parameters will be fitted against all bands.

In the second iteration the $p-$parameters are adopted from the first fitting. From that point, the band specific $d$ parameters are



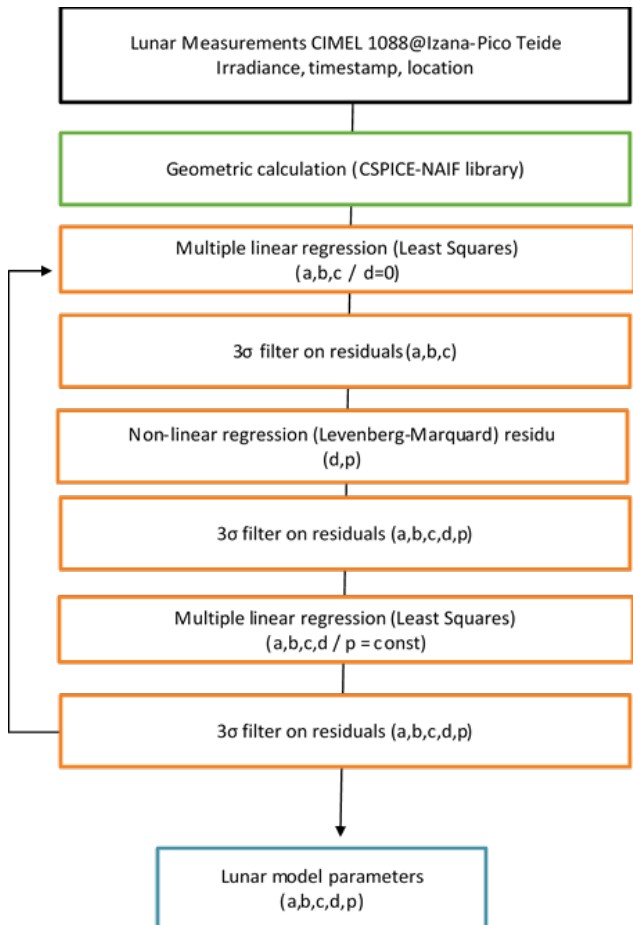

**Figure 8.** Scheme of the lunar model coefficients regression algorithm.

re-fitted in a linear least squares with all $a$, $b$ and $c$ parameters. The $p$ parameters are then used in further regression and outlier removals. Finally, again a full linear fitting is performed on the entire equation, keeping the previously derived non-linear parameters constant ($p-$parameters). It is important to note that a 3-sigma outlier removal is applied after all regression steps to ensure the quality of the whole fitting analysis.

## 6.3 Uncertainty analysis

The lunar model fit is a multi-step process as described in section 6.2, where the linear part of the model is fit for each band, outlier removed, then the non-linear part is fit. This is followed by further outlier removal and finally the linear part is fit again. The whole multi-step process is itself iterated.

The approach to uncertainty analysis in the model regression follows the Monte Carlo Uncertainty Analysis (MCUA) methods outlined in the GUM (JCGM101, 2008). The Monte Carlo analysis is fed with random errors, based on the knowledge of





|  | 440 nm | 500 nm | 675 nm | 870 nm | 1020 nm | 1640 nm |
|---|---|---|---|---|---|---|
| S | 0.77% | 0.73% | 0.55% | 0.63% | 0.31% | 0.31% |
| C | 1.1% | 1.1% | 1.1% | 1.1% | 1.1% | 1.1% |

**Table 9.** Systematic uncertainties per band $S_\lambda$ and to all measurements $C$

uncertainties provided with the measurements. In practice, every measurement is slightly adjusted with a random error which lies within the uncertainty interval. The MCUA process is based on a measurement model. The input irradiance values (the TOA irradiance values for each night obtained by the Langley method) are treated as:

$$E_{i,\lambda} = E_{i,\lambda}^{True} \cdot (1 + R_{i,\lambda})(1 + S_\lambda)(1 + C) \tag{16}$$

Where, $E_{i,\lambda}^{True}$ is the nominal "true" value for the TOA irradiance in spectral band for the -th observation; $R_{i,\lambda}$ is the error in the observation in spectral band for the th observation due to random effects, expressed in relative terms; $S_\lambda$ is the error in the observation that is common for all measurements in this band, expressed in relative terms; $C$ is the error in the observation that is common for all measurements in all bands, expressed in relative terms. The error values are unknown; but are drawn from a probability distribution with a standard deviation given by the relative uncertainty associated with this effect and with 505 an expectation value (central value) of zero.

$R_{i,\lambda}$ takes a different value for every observation. This comes from random processes relating to the measurement of the TOA irradiance for a particular night. These include instrument noise, instrument temperature changes and atmospheric changes, and relates to the relative uncertainty in the Langley Plot intercept.

$S_\lambda$ takes the same value for every observation for a single spectral band. This comes from effects that are common for that 510 band – and mostly that is from the NPL calibration of the instrument. Any uncertainty associated with the NPL calibration is "fixed" into that calibration and applied to all measurements.

$C$ takes the same value for every single observation in all spectral bands. This comes from effects in the NPL calibration that are wavelength independent, e.g. from a distance offset on an instrument alignment.

The consideration of error correlation structures in the calibration and measurement uncertainties informs the systematic 515 uncertainties used in the MCUA. The uncertainties associated with random errors is given by the uncertainty in the Langley intercepts as described in section 5.3.2 (Table 8).

The MCUA is performed only for the final iterative step in the model regression. The fit routine is run 1000 times. For each iteration a single value of the error $C$ is drawn randomly from a Gaussian distribution with a central value 0 and a standard deviation equal to the uncertainty associated with $C$. 6 values $S_\lambda$ are used each corresponding to a different spectral band, and 520 as many values of $R_{i,\lambda}$ are used as are needed, the number of spectral bands multiplied by the number of observations.

Conceptually, the input values are altered by these errors, the fit is performed, and a model is derived based on those errors. This is repeated 1000 times to give 1000 different models. These model outputs are then used to determine the uncertainty



**Figure 9.** Uncertainty levels determined from MCUA for the Cimel photometer spectral bands

associated with the model, by considering the uncertainty associated with each of the parameters and the covariance between them. This is done statistically, using the standard deviation of the 1000 instances of each fit parameter.

Figures 9 a-f show the uncertainty levels for all bands averaged per 5-degree phase angle bins. Uncertainty levels at 95.5% ($k = 2$) and 99.7% ($k = 3$) confidence levels are shown as well as the mean $E_{i,\lambda}$ obtained over the 1000 model perturbations. At the 95.5% confidence level all bands perform well below 2% except for the 440 nm band. For 99.7% confidence, all bands perform at approximately 2.5% uncertainty except for 440 nm and 500 nm bands.



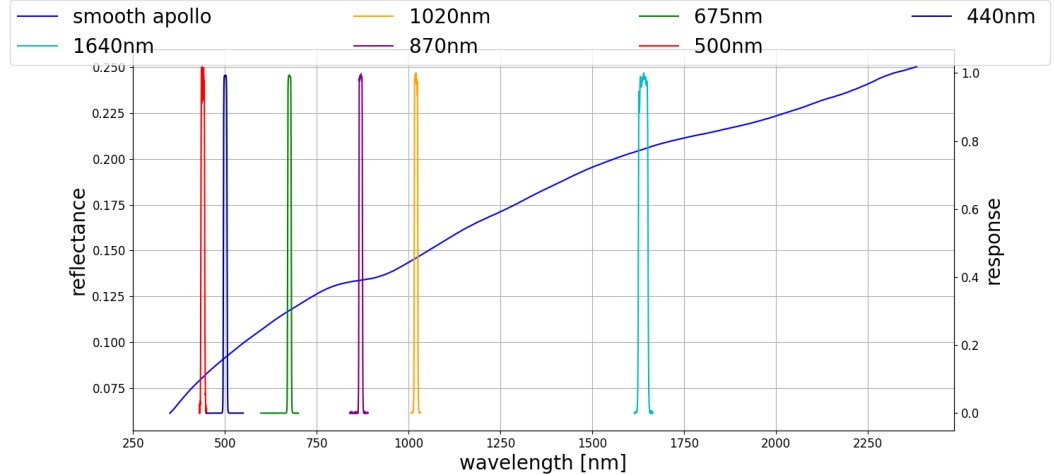

**Figure 10.** Cimel response curves and interpolated smoothed reflectance.

## 6.4 Spectral Interpolation

A reflectance spectrum of the Moon is used to increase the model spectral resolution. The lunar model calculates reflectance at the 6 Cimel photometer bands. The spectral range of the model spans from 440 nm to 1640 nm, in discrete wavelength positions (Figure 10). Therefore, in intermediate model regions, the spectrum needs to be adjusted and reconstructed.

The first step is the smoothing of the measured spectrum with a reference reflectance. Reflectance profiles of two Apollo 16 lunar probe samples are used to construct the reference reflectance spectrum (Kieffer and Stone, 2005). This spectrum was
used to radiometrically rescale and interpolate the ROLO model output at the ROLO measurement spectral bands. The resulting reflectance $R_{mix,\lambda}$ is a linear combination of both spectral ($\lambda$) reflectances, $R_{breccia,\lambda}$ and $R_{soil,\lambda}$ measured for breccia and soil samples.

$$R_{mix,\lambda} = 0.05 \cdot R_{breccia,\lambda} + 0.95 \cdot R_{soil,\lambda} \tag{17}$$

The mixed reflectance is derived for every lunar model wavelength. These values are used to calculate the least absolute
deviation regression values with respect to the reflectance obtained at the Cimel bands. The lunar model reflectances used in the regression are calculated for every measurement specifically. This regression results in a set of smoothing coefficients, which are applied to the spectral reflectance model, resulting in a smoothed lunar reflectance spectrum.



## 6.5 LIME comparisons

The LIME model outputs have been compared with the satellite spectral imagers PROBA-V and PLEIADES-HR-1B. Lunar
acquisitions from PROBA-V are limited in the range of lunar phase angles but cover an extensive time period. Acquisitions
from Pleiades are limited in time, but cover more of the lunar phase.

### 6.5.1 Comparison Methodology

The sensor irradiance measurements $E_k$ of the Moon at $k$ spectral band are compared to LIME by defining the radiometric
ratio between the instrument (measurement, $E_{k,meas}$) and model outputs ($E_{k,model}$) with the simple relationship:

$$
\quad C_k = \frac{E_{k,meas}}{E_{k,model}} - 1 \tag{18}
$$

The LIME model provides the lunar irradiance for a given viewing geometry and spectral response and, as such, there are
some limitations in the comparison which must be taken in to account. Lunar phase angles are limited to between 2 and 90
degrees, and at present the model covers the spectral range of 400 nm to 2500 nm.

Each Earth Observation sensor lunar acquisition feeds a minimum set of input parameters to the model for the comparison,
including timestamp of acquisition [Julian Day], position of sensor [J2000 co-ordinates], and integrated irradiance from lunar
acquisition.

The model calculates the geometric parameters per acquisition required for the comparison, i.e. phase angle, solar seleno-
graphic longitude, observer selenographic latitude and longitude and distances between Sun, Moon and observer using the
NASA SPICE toolkit. The instrument spectral response curve is used to calculate the model irradiance. As explained in the
previous section, the LIME model is generated only at the wavelengths of the Cimel instrument and these model outputs are
spectrally interpolated at present using the Apollo rock samples reflectance, as in the ROLO model.

Figure 11 is a flowchart of the procedure that is applied to the model input. The output of the procedure is the simulated lunar
irradiance, which can be compared with the correlated measured irradiance. The different steps that are applied to obtain the
simulated sensor irradiance are: calculation of the geometry (using SPICE); calculation of the model reflectance for all model
wavelengths; spectral adjustment, by taking into account the sensor spectral response; conversion of reflectance spectrum to
irradiance spectrum; integration with sensor response curve; and correction for the distance factor of the input irradiance value.
Finally, the obtained modelled irradiance can be compared with the correlated sensor lunar measurement.

### 6.5.2 PROBA-V Results

The PROBA-V instrument is a multi-spectral imager with four broad spectral bands: BLUE, RED NIR and SWIR, centered at
450, 645, 834 and 1665 nm. PROBA-V lunar images are acquired twice every month, approx. 7 degrees phase angle before
and after full Moon. Since the beginning of the launch, the Moon has been recorded. To prepare the L1A PROBA-V data for
comparison with the lunar model, several processing steps are required. The first one is to find all Moon-pixels in the image



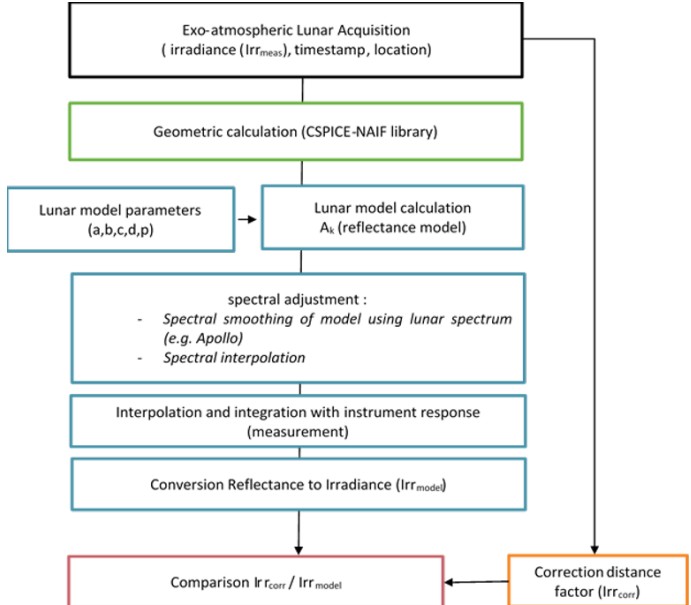

**Figure 11.** Measurement and Model comparison procedure.

| BAND | Blue (450 nm) | Red (645 nm) | NIR (834 nm) | SWIR (1665 nm) |
|---|---|---|---|---|
| AVG [%] | -0.7 | -0.0 | 1.2 | 25.2 |
| STDEV | 0.9 | 0.6 | 0.7 | 0.4 |

**Table 10.** PROBA-V comparison to LIME: mean difference (in %) and standard deviation for each spectral band.

(masking). Then locate the center row of the Moon and get the exact timestamp and satellite position (in J2000-coordinates) for this central row. Third, convert Moon-pixels into radiance (apply instrument calibration parameters). Four, integrate all
Moon-pixels. Last, calculate the solid angle per pixel to finally derive Moon irradiance.

The comparison between LIME and PROBA-V for each sensor spectral band are given in Figure 12, with the mean differences summarized in Table 10. As it can be seen, the comparison results in differences that are generally within ±2% except in the SWIR channel, where they are as high as 25%. It is clear that the PROBA-V SWIR data need further investigation. Critical step for the processing is masking, which appears to be rather difficult for the noisier SWIR channel. The masking is the basis
for all further processing and therefore the SWIR absolute level of the lunar irradiance should be assumed immature.

A limited analysis is done, to evaluate trending capabilities of the LIME model. The ROLO lunar model has been applied in the past to evaluate possible instrument degradation. As mentioned, the Moon has high reflectance/irradiance stability over time and consequently yearly trends of 1% can be detected with sensor lunar acquisitions. PROBA-V has monthly lunar data over +5 years, therefore it is a good data set to check the trending capabilities of the lunar model. The linear regression trends for the
differences between PROBA-V and LIME have been calculated (see equations in Figure 12). These trends are cross checked



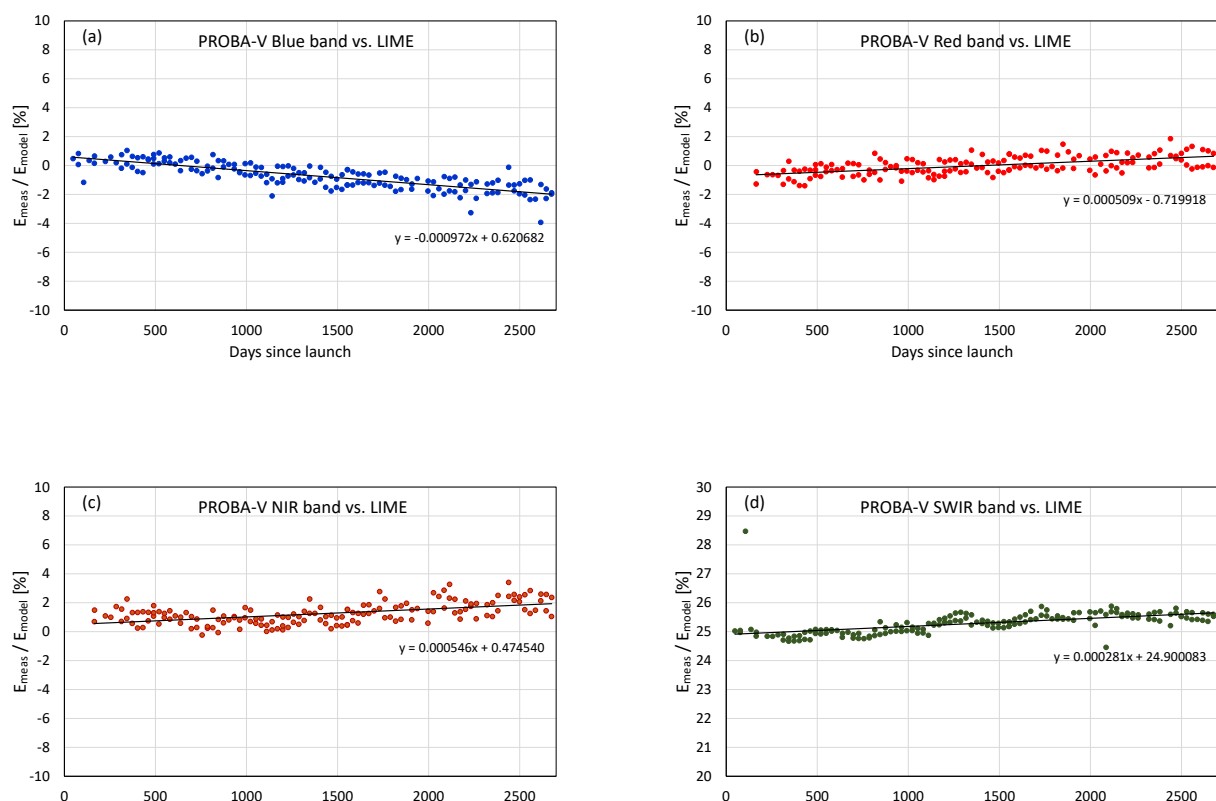

**Figure 12.** PROBA-V irradiance compared to LIME: differences in % since launch of the sensor. (a) Blue band; (b) Red band; (c) Near infrared (NIR) band; (d) Short-wave infrared (SWIR) band.

and confirmed by application of other methods to PROBA-V sensor data, like Pseudo-invariant Calibration Sites (PICS) desert Libya-4 (Sterckx et al., 2014).

### 6.5.3 Pleiades-1B Results

The Pleiades-1B HR imaging instrument (also called PHR1B) is a high resolution multi-spectral imager. It has five spectral
bands in the VNIR region: Blue (430-550 nm); Green (490-610 nm); Red (600-720 nm); Near Infrared (750-950 nm); and Panchromatic band (480-830 nm).

In total 68 lunar observations are considered in this study, spanning the period between 18-Feb-2013 until 07-Apr-2017. The measurements are a combination of 2 campaigns in Feb/2013 and Mar/2013 recording at sparse lunar phase angles over the entire cycle. These are supplemented with routine observation around 40 degrees phase angle acquired every few months for



several years. Even if the measurements are sparse with respect to the lunar phase angle, they cover a considerably wider range of phase angles than the PROBA-V observations.

Similarly to the PROBA-V analysis above, we have generated a model output for all Pleiades observations and spectral bands. The mean difference (in %) has been calculated and is provided in Table 11. The output of the LIME model irradiance is slightly lower than the PLEIADES irradiance levels, calibrated with other vicarious calibration methods. The comparison shows differences below 5% for the visible spectral bands and above 6% for the NIR and PAN channels.

| BAND | Blue | Green | Red | NIR | PAN |
|---|---|---|---|---|---|
| AVG [%] | 3.2 | 4.7 | 4.5 | 6.8 | 6.1 |
| STDEV | 1.4 | 1.1 | 0.9 | 1.1 | 9.6 |

**Table 11.** Pleiades-1B comparison to LIME: mean difference (in %) and standard deviation for each spectral band.

### 6.5.4   Comparison to GIRO

The LIME model has been also been compared to the VITO implementation of the ROLO model, i.e. the GIRO. The first results indicate that LIME predicts 3% - 5% higher disk integrated lunar irradiance than the ROLO/GIRO model for the visible and near-infrared channels, the difference being smaller for longer wavelengths. The comparison exercise was done by simulating model outputs for the PROBA-V spectral bands. An overview of the comparison is given in Table 12.

Moreover, the LIME output was compared with Sentinel 3B measurements. The calculated instrument irradiances were within the 2% uncertainty range for most of the bands (Neneman et al., 2020). For further details about all the mentioned comparisons, please visit https://calvalportal.ceos.org/lime-documents.

### 7   Degree of Linear Polarisation

The instrument used to measure lunar irradiance (Cimel CE318-TP9) allows for characterisation of portion of polarized light from the Moon. The construction of the Cimel instrument prevents the Stokes parameters from being measured directly, but the degree of linear polarisation (DoLP) can be calculated from the different instrument filter outputs. This implies however, that it is not possible to measure negative polarisation. As a first approach for LIME, all measurements below the inversion angle of 23 degrees phase angle (Shkuratov et al., 2015) are set to negative. This is a pragmatic approach to use the negative

| BAND | Blue (450 nm) | Red (645 nm) | NIR (834 nm) | SWIR (1665 nm) |
|---|---|---|---|---|
| AVG [%] | 4.4 | 3.4 | 3.2 | 3.0 |
| STDEV | 1.4 | 1.0 | 0.9 | 1.1 |

**Table 12.** Comparison of GIRO-ROLO to LIME computed for the PROBA-V: mean difference (in %) and standard deviation for each spectral band.



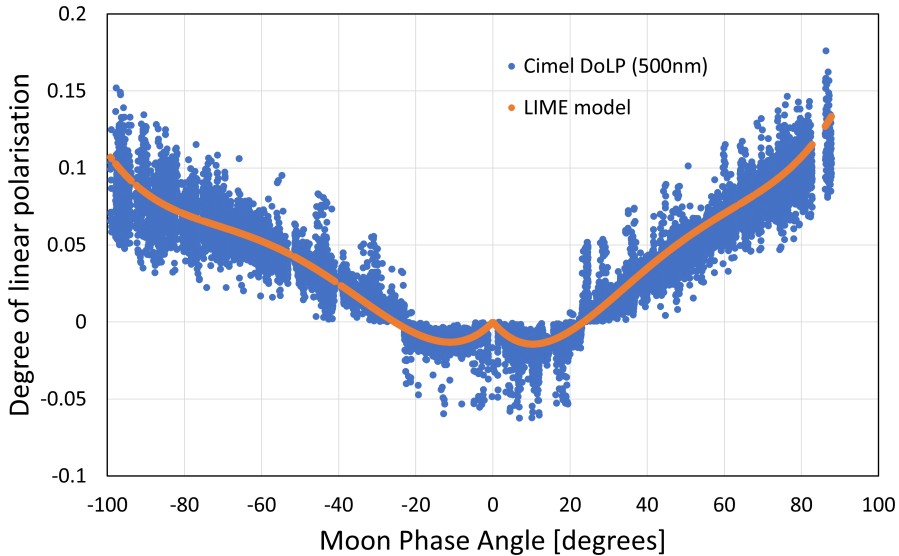

**Figure 13.** DoLP measurements and curve fitted DoLP for the 500 nm band.

solution of the DoLP formula. The way to convert the output of the instrument to Stokes parameters, or a way to calculate a negative solution of the DoLP formula, is currently under investigation.

In the Cimel photometer, three linear polarized filters are oriented 60º from each other, measuring directly the raw polarized signals. The three filters give a value for $S_{p1}$ , $S_{p2}$ , $S_{p3}$. The degree of polarisation is derived with the following formula (Li et al., 2010):

$$DoLP = \frac{2\eta\sqrt{S_{p1}^2 + R_{12}^2 S_{p2}^2 + R_{13}^2 S_{p3}^2 - R_{12} S_{p1} S_{p2} - R_{13} S_{p1} S_{p3} - R_{12} R_{13} S_{p2} S_{p3}}}{S_{p1} + R_{12} S_{p2} + R_{13} S_{p3}} \tag{19}$$


$R_{12}$, $R_{13}$ are the corrections for total polarisation transmittance and $\eta$ is the polarisation calibration coefficient. These were calculated during the calibration of the instrument.

All measurements performed by the #1088 instrument also include polarized Moon irradiances. This means for the period of about 1 year, more than 120000 measurements of lunar polarized light are available. Not all measurements are done at full

night-time and these need to be filtered from the regression. The measurements are filtered on time – between 23h at night and 2h in the morning and outliers are removed (i.e. cloud contaminated measurements). Measurements with negative and positive phase angles are split to be able to produce a separate regression on both sides. About 25000 measurements per phase sign are used to perform the model regression. The spectral bands are treated separately.

The model is limited to be between 0° and 90° absolute phase angle. All polarisation measurements outside these angles are

removed from the regression of DoLP curves.





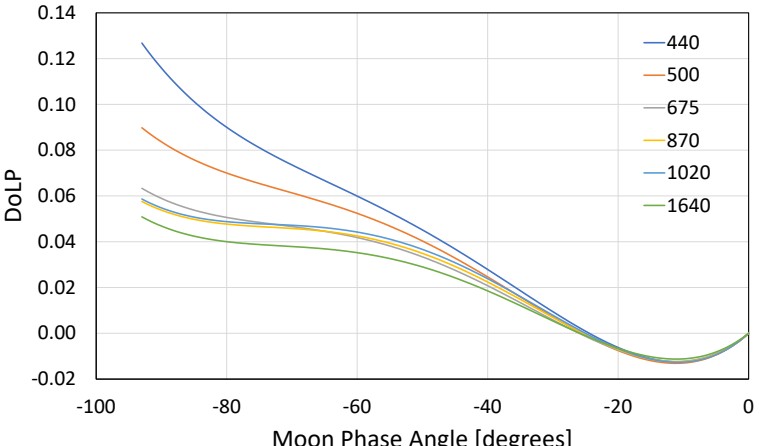

**Figure 14.** Modelled DoLP for negative phase angles for the LIME wavelengths (nm).

As can be observed in Figure 13, the DoLP can be modelled using a fourth order polynomial with the intercept set to zero. From this polynomial the DoLP value is calculated directly:

$$DOLP = a_1 \cdot g + a_2 \cdot g^2 + a_3 \cdot g^3 + a4 \cdot g^4 \tag{20}$$

where $g$ is the phase angle in degrees and $a_i$ are the fitting coefficients. The modeled DoLP for all spectral bands is given in
Figure 14, where only negative phase angles are shown. The spectral DoLP as a function of the phase angle can be considered as a preliminary additional output of the LIME model, although the uncertainty estimation for this physical quantity has not been developed yet. Moreover, the results need to the compared to the previous observations reported in the literature (e.g. Lyot, 1929; Shkuratov et al., 2008).

## 8 Conclusion and Future Development

The measurements are then fitted to the ROLO equations to obtain the set of coefficients needed to provide lunar reflectance. The Cimel photometer was calibrated so that the irradiance measurements are directly traceable to the SI. A rigorous uncertainty analysis has been performed, from instrument calibration to individual observations, to the model output. It indicates that the model output uncertainty is below 2% ($k = 2$). The comparison with satellite lunar acquisitions, generally confirms this estimation, although further investigations are needed concerning the PROBA-V SWIR channel. LIME outputs are 3% -



5% higher than the GIRO-ROLO model for the visible and near-infrared channels. The LIME model performance could be further tested if used for derivation of aerosol optical depth. The lunar observations that are the base for the current LIME model span from 2018 to 2022. More data are still needed to cover the full range of selenographic latitude and longitude using our observation strategy and is essential for a complete lunar irradiance model.

A new lunar model, the Lunar Irradiance Model of ESA (LIME), has been developed. The strategy for deriving the model
involved using a robust filter radiometer, the Cimel CE318, extensively used in the AERONET network, and conducting direct Moon observations at varying Moon zenith angles. These observations are suitable for deriving the Top-of-Atmosphere lunar irradiance using the Langley plot method, provided that the measurements are acquired at high-altitude stations such as Izaña and Teide Peak in Tenerife, Spain, where the atmospheric transmission changes minimally over the Langley period (approximately 1.5 hours per night). The obtained measurements are then fitted to the ROLO equations to obtain the set of
coefficients necessary for determining lunar reflectance. The lunar photometer was calibrated, ensuring that the irradiance measurements are directly traceable to the International System of Units (SI). A comprehensive uncertainty analysis has been conducted, encompassing instrument calibration, individual observations, and the model output. The analysis indicates that the model output uncertainty is below 2% ($k = 2$). When comparing the model to satellite lunar acquisitions, this estimation is generally confirmed, although further investigations are required regarding the PROBA-V SWIR channel. LIME outputs
show a 3% - 5% higher value than the GIRO-ROLO model for the visible and near-infrared channels. The performance of the LIME model could be further tested if employed for deriving aerosol optical depth. The lunar photometer also retrieves the degree of linear polarisation of the Moon light, that has been fitted as a function of the phase angle and included as an added value of the LIME output. Further investigations are needed to provide an uncertainty estimation to this magnitude. The lunar observations forming the basis of the current LIME model span from 2018 to 2022. Additional data are still required to
cover the complete range of selenographic latitude and longitude using our observation strategy. Such data are essential for establishing a comprehensive lunar irradiance model.

*Author contributions.*  calibration of CIMEL. S.T, E.W, C.T, A. Be., A.Ba., A. Bi.; instrument Operation and data processing A.Be, A.Ba, R.G.; model derivation and intercomparison S.A; conceptualisation of uncertainty analysis E.W and S.T; writing C.T., R.G., S.T, S.A, A.Ba., M.B. – review and editing all.

*Acknowledgements.*  We thank Aimé Meygret at the Centre National d'Etudes Spatiales for providing the Pléïades data used in this study. Thanks to the staff of Observatorio Atmosférico de Izaña (AEMET) for the photometer operation at Teide Peak and Izaña, and for ensuring the high quality of data by intensive data monitoring. This work has been developed within the framework of the activities of the World Meteorological Organization (WMO) Commission for Instruments and Methods of Observations (CIMO) Izaña Testbed for Aerosols and Water Vapour Remote Sensing Instruments. AERONET sun photometers at Izaña were calibrated through the AEROSPAIN Central Facility
(https://aerospain.aemet.es/) and supported by the European Community Research Infrastructure Action under the ACTRIS-IMP (grant agreement no. 871115).



*Competing interests.* The authors declare no conflict of interest.



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
