# Peer review of "LIME: Lunar Irradiance Model of ESA, a new tool for the absolute radiometric calibration using the Moon"

_EGUsphere, 2023_

## Author Comment (AC1)

Answer to referee RC1

Dear referee, many thanks for the constructive comments and useful technical corrections. The responses are given below.

1. general comments

This paper outlines very well the strategy used to develop a lunar irradiance model from new ground-based measurements obtained from a high altitude location, as aneed shown from literature overview to develop an SI-traceable absolute irradiance model of the Moon.

LIME, the Lunar Irradiance Model of the European Space Agency, is a new lunar irradiance model developed from ground-based observations acquired using a lunarphotometer operating from the Izaña Atmospheric Observatory and Teide Peak, perfect sites for such measurements. A key attribute of the LIME model is a rigorousuncertainty analysis and the ambitious target of a sub-2% uncertainty in the resultant model as it is fi nally proved by this study.

This new model is expected to play an important role on EO radiometric calibration, which can be validated using radiometrically stable natural targets, like the lunardisk irradiance. With this information, Earth Observation measurements can be radiometrically linked to all past, present and future sensors having performed similarmeasurements.

Strategy for extraterrestrial Moon irradiance retrieval is described and easy to follow., along with the calibration. The linearity of the measurements with CIMELphotometer is tested. Also the thermal sensitivity of the instrument and the irradiance responsivity of each lunar photometer spectral channel were assessed. Rigorousuncertainty analysis of the calibration was performed, sources of uncertainty for individual lunar observations were correctly identifi ed and assessed.

The LIME model as derived from the lunar irradiance measurements from the Cimel photometer is fully described. LIME basically improves ROLO model by using anindependent set of C-coeffi cients for each spectral band of the model, calculated using diff erent steps in the procedure (Figure 8 in the article).

A refl ectance spectrum of the Moon is used to increase the model spectral resolution.

The LIME model outputs have been compared with the satellite spectral imagers PROBA-V and PLEIADES-HR-1B but considering the limitations of such comparisons.Procedure nicely and clear described and also represented in Figure 11

The manuscript is overall well written and addresses globally relevant issues.

2. specific comments

This paper is a response to a need related to the calibration of Earth observation sensors (to ensure the continuity of long-term and global climate records) and especially important for satellite sensors, calibrated prior to launch because their susceptibility to degradation in space.

I think the authors should add a short paragraph in the conclusions related to availability of LIME to be used by other interested parties.

Indeed we are working in a web tool with open access for the scientific community to run LIME and use it for research purposes. The tool is not ready yet but could be released before the end of 2023.

We have added the following sentence: "The European Space Agency plans to provide an open web tool for the LIME model to be used for research purposes."

3.technical corrections

You are using VITO in the abstract- Please consider to write the name of the Institute here in parenthesis

Done.

Line 465- equation 15- please use subscript for 3 and 4 of the coeffi cients "p3" and "p4"

Done

Figure 8- please change Pico Teide with the English version-Peak Teide –

Done

Line 519 (page 25) please write "six values of S..." instead of "6 values of S..."

Done.

Equation no.20 (page 33) Please make the correction: for fi tting coeffi cient "a4"- 4 should be subscript

Done.

In Figure 14- for a better visualisation in the graph the authors should consider black color for 440 nm or 1020 nm (since now it could be diffi cult to distinguishbetween the two)

Done.

---

## Author Comment (AC2)

Answer to referee RC2

Dear Tom, many thanks for the constructive comments and useful technical corrections. The responses are given below. The dataset of figures 6 and 7 has been provided to the editor.

This paper is a welcome contribution to the fi eld of lunar calibration. The LIME project represents a significant eff ort toward producing a more accurate model for the exo-atmospheric lunar irradiance, and there has been a need to consolidate the documentation on LIME development. This paper covers the topic comprehensively, with including uncertainty analyses for each component of the model development. There are a few technical points that should be explained more fully, noted in specific comments below. The manuscript is well written and is recommended for publication.

Specific Comments

Section 2.3

A more up-to-date accounting of the uncertainty in the ROLO model is given in the reference Stone et al. 2020, cited elsewhere in this manuscript. But it generally aligns with the 5-10% value quoted here, so this is a very minor issue.

Stone et al. 2020 reference has been added to this paragraph.

Section 3.5

It would be informative to have values for: 1) the typical duration of the Langley acquisition period, 2) the typical lunar phase change during this period (i.e. quantify"minute changes"), and 3) the numerical range of the correction A(t_ref,lambda)/A(t,lambda).

A Langley acquisition at the latitude of Izaña, lasts about 01:35 to 02:15 h depending on the season. The air mass changes rapidly from 7 to 2, i.e., solar elevations from approximately 8 to 30 degrees. Just for comparison, at 37º latitude, the time in winter to change from air mass 7 to 2 is more than 3 h. At higher latitudes, air mass 2 is not reached in winter. Therefore in a typical Langley sequence the phase angle change is about 1 degree and the mean A(t_ref,lambda)/A(t,lambda) is as small as 1.0006. This is provided in Table 5 of the manuscript.

Section 4.1

It would be informative to have typical DN values expected for Full Moon and for 90 degrees phase, to see where the lunar measurements fall on the linearity factor plots in Fig.4.

The raw signal in 440nm channel (worst case) range from 200 up to 4500 counts between 90 degrees and full moon. At 1640nm (channel with highest signal), it ranges from 3000 to 20000 counts. See below the raw signal for 440nm (channel 4) during March 2023 moon cycle (1st to 3rd quarter). Note, however, that we also intend to use solar Langley plots to monitor the instrument stability (and eventually derive reflectance from Moon/Sun top of atmosphere signals derived from Langley plots). That's why we tested linearity in a broad range. Solar observations have signals about 1E05 at noon. We added this information to the text:

"A linearity factor was calculated for illumination levels varied by changing neutral density filters, covering the typical signal range from Moon measurements at high phase angles (about

200-3000 counts depending on wavelength) up to direct Sun measurements near noon (about 1E05 counts). Then the results were averaged…[]"

[Figure]

Section 4.2

The much larger temperature sensitivity for the Si 1020 channel suggests using the InGaAs 1020 channel - are there other considerations that led to the choice to use the Si channel, despite the potential added uncertainty?

The 1020-Silicon channel is the one used operationally in AERONET, for several reasons, mainly that it is the spectral channel acquired first in the Sun/Moon observations and that it has been extensively characterized, including the temperature dependence, which is made individually for each instrument. With this choice we guarantee the consistency with the other wavelengths. The 1020-InGaAs is a redundant measurement used for quality control purposes only (detection of obstructions in the optical path).

Section 4.4

Units should be given for the calibration coefficients, preferably in Table 4 but alternatively in the text.

Added to table 4, thank you.

Section 5.1

The text seems to imply that the CAELIS software conducts the uncertainty analysis described in section 5.3 - is that correct, or is this analysis done offline from CAELIS? How are the "statistical indicators of the fit quality" used?

The uncertainty analysis is done offline CAELIS. In CAELIS only the Langley plots are performed and the initial value of the uncertainty of the intercept is calculated. This value can be later modified by the procedure described in section 5.3.2. This has been clarified in the text.

Section 5.3.1

What is the relation between count signal D(lambda,t) and V^m(lambda,t) used in the Langley plots and converted to irradiance through the calibration coefficient, as inEq.10?

There was a typo, it must be $V(\lambda,t)$ in this section and also in section 5.3.2.

Please provide the method used to compute the lunar zenith angle (theta) using the SPICE tools, as mentioned in section 5.1. Kasten and Young (1989) state:"substantial errors will be

incurred if unrefracted elevations are used" when applying their Eq.3, from which Eq.14 here is derived (see also the Technical Correction below). This potential source of uncertainty in theta propagates into an uncertainty in the airmass values m, and the assumption that uncertainty associated with airmass is negligible might not be valid if refraction is not taken into account for specifying the zenith angle.

This refraction correction is not applied. For airmass <5 the use of refraction is negligible (elevation is >11 degrees, refraction<4 arc minutes for Izaña/Teide pressure, airmass change 0.02). And for higher elevations it decreases rapidly. For the calculation of the model parameters this is not relevant.

Presumably the choice to restrict the airmass range to 2-5 is also driven by the time required to collect the Langley data, to limit the effects of variability in the atmosphere.

Correct. The limitation to 2-5 in the airmass range reduces the time needed for the Langley acquisition as well as the uncertainty associated to airmass, because the sentence above by Kasten applies to large zenith angles (airmass larger than 7). For shorter zenith angles the airmass uncertainty can be considered negligible, as we assume in the uncertainty budget estimation.

Section 5.3.2

What is the "expected chi^2" value? What was the rationale for choosing this value?

The $\chi^2$ test is used to validate statistically the model of the straight line fit. The test checks if the spread of the data is consisted with its uncertainties. The expected $\chi^2_\nu$ is derived as $\chi^2$ distribution for the degrees of freedom per Langley plot, thus the individual values depend on how many measurements were used for the fit. We followed the methodology outlined in ISO/TS 28037:2010, "Determination and use of straight-line calibration functions" to validate the model and its uncertainty estimates.

The critical value of the $\chi^2$ test results from the degrees of freedom and the selected level of significance. Critical values can be found in a table of probabilities for the $\chi^2$ distribution. If the observed $\chi^2$ in the data is large than this critical (expected value), we reject the initial uncertainty of the intercept. The uncertainty on each data point needs to be increased until the $\chi^2$ test is passed, and then we calculate a new uncertainty of the intercept.

The text has been slightly modified with these clarifications and the reference to ISO/TS 28037:2010 has been added.

The uncertainties in the Langley plot intercept u(ln[V_0]) are critical, since the V_0 quantities from the fits ultimately are used for deriving the model coefficients in section 6.2, and the uncertainties in V_0 are central to the model uncertainty analysis in section 6.3. The quoted uncertainties u(ln[V_0]) in Figs.6 and 7 seem unreasonably small for fitted parameter uncertainties (specifically, the intercept) derived from these linear regressions, given the range of airmass and the extrapolation to zero airmass. But these parameter uncertainty values can't be checked without having access to the Langley datasets. This reviewer is willing to run checks on these fi t results if the authors would provide the data for Figs.6 and 7, with including the measurement uncertainties.

The data for Figs.6 and 7 have been provided via email. There was also a mistake in Table 7, where the signal uncertainty is provided (*D* variable in the first version, now *V*). They were indeed too small (by mistake), by a factor of 10. This is now corrected.

Section 6.2

This section is missing a description of how the geometry variables g, Phi, theta and phi are computed for each data point.

The calculation of the observer, lunar and solar geometries is done by the application of the CSPICE library, provided to the public by NASA NAIF. This information is now given in the text.

References

Acton, C.H.; "Ancillary Data Services of NASA's Navigation and Ancillary Information Facility;" Planetary and Space Science, Vol. 44, No. 1, pp. 65-70, 1996. DOI 10.1016/0032-0633(95)00107-7

Charles Acton, Nathaniel Bachman, Boris Semenov, Edward Wright; A look toward the future in the handling of space science mission geometry; Planetary and Space Science (2017); DOI 10.1016/j.pss.2017.02.013

Web location :

https://naif.jpl.nasa.gov/naif/

For the non-linear part of the fitting to determine the p coefficients, how is the degeneracy of the d_1 and d_2 terms handled? The form of these two terms is identical in Eq.15.

The non-linear part of the model is treated in 2 parts, where the d_1 and d_2 terms are fitted after the p-terms. The values of the d parameters are fitted in the second iteration of the model fit, where p-terms are kept from the first iteration and a full refit (a,b,c,d) terms is done by linear Levenbergh Marquardt fit.

Section 6.3

This section is very important, because it presents the method used to derive the uncertainty in the LIME outputs. More detail here would be helpful, as suggested in the following:

Presumably the values in Table 9 are used to specify the variances of the Probability Distribution Functions (PDFs) for S_lambda and C; this is not mentioned in the text. How are these table values related to the instrument calibration uncertainties given in section 4.4, Tables 3 and 4? Is the PDF for random errors R_i,lambda constructed from the individual measurement errors, or is it an analytic function (such as a Gaussian) with characteristic parameters specified by the collective results given in section 5.3.2 and Table 8?

The values in the table are derived from the lab calibration setup and are the synthesis values for system and band specific uncertainties. These values are used, together with the measurement Langley uncertainties (eq. 6) to perturbate the measured lunar irradiance. The R (measurement uncertainty), C and S combined give one overall uncertainty value (% of the signal) for a certain measurement. The input to the PDF is this value. The distribution of the error is assumed to be gaussian and thus applied to the input measurement.

Additional details are needed to explain the statement: "These [1000] model outputs are then used to determine the uncertainty associated with the model, by considering the uncertainty associated with each of the parameters and the covariance between them." How are the uncertainties in the (18) model parameters combined to give the uncertainty distributions for the full model for each band? Presumably the latter are the distributions from which the confidence intervals shown in Fig.9 are derived, which are considered specifications of the uncertainty in the LIME outputs.

The models derived from the perturbation of the lunar irradiance measurements (1000 iterations) allow for the calculation of the model uncertainty budget, by the calculation of the covariance matrix for all model parameters for all bands together.

Lastly, it would informative to see values for the mean absolute residuals of the full dataset, between the TOA irradiance measurements and the corresponding LIME outputs. These are an indicator of the relative accuracy of the model, and they would provide a useful comparison to the results from the Monte Carlo analysis.

This is an example of residuals for 440 and 870 nm bands between model and CIMEL (m21xls)

[Figure]

Section 6.4

It would be informative to see in Fig.10 the model outputs for the 6 Cimel bands and the smoothed lunar reflectance spectrum for an example measurement.

The next plot contains the output of the model reflectance and the smoothed apollo16 reflectance

[Figure]

Case :

| | |
|---|---|
| sun_sel_lon | 33.17843893 |
| obs_sel_lat | -2.096516 |
| obs_sel_lon | 2.175489 |
| phase_angle | -30.9993085 |
| distance_sun_moon | 149664930.2 |
| distance_obs_moon | 369123.6044 |

What is the form of the regression that is solved using least absolute deviation?

This is essentially a least squares regression, by fitting a line through the different spectral points to estimate a and b (with y=ax+b). The first attempt is done by chi-square minimisation, to find initial guesses. The minimisation is based on the median of all absolute deviations.

Section 6.5.4

In the lunar calibration field, the common usage of "GIRO" means "GSICS Implementation of ROLO", i.e. the lunar model software built by EUMETSAT and validated against the USGS ROLO model. It is confusing to refer to the VITO implementation of the ROLO model as GIRO.

Changed to "GSICS Implementation of ROLO".

It would be worth mentioning that the 3% to 5% differences between the irradiances predicted by ROLO/GIRO and LIME are within the range of uncertainties quoted for the two models.

The LIME documents on the CEOS calval portal at URL provided were not accessible on 10 August 2023 - the site gives the message: "Error: You do not have the required permissions."

An issue with the website has been solved.

Section 7

With due respect for the authors' discretion as to the paper's contents, may I suggest to consider removing this section. This topic could support a separate paper, the text indicates that the study is incomplete, and the material on irradiance measurements, modeling, and uncertainties makes a very substantial paper on its own.

We have agreed to follow this suggestion and remove the section devoted to polarization.

Technical Corrections

Section 5.3.1: m(Theta) before Eq.12 should be lower case theta.

Corrected.

The form of Eq.14 is incorrect as written - the constants from Kasten and Young (1989) have been incorrectly converted from degree to radian measure.

Corrected.

Section 6.3: several instances of misprinted "i-th observation"

Corrected.

Section 8: this section might actually start with the current second paragraph - the first paragraph appears to be mistakenly present.

Corrected.